

# Aerosol radiative effects and feedbacks on boundary layer meteorology and PM₂.₅ chemical components during winter haze events over the Beijing-Tianjin-Hebei region

Jiawei Li [1], Zhiwei Han[*,1,2], Yunfei Wu[1], Zhe Xiong[1], Xiangao Xia[3], Jie Li[1,2], Lin Liang[1,2], Renjian Zhang[1]

[1] Key Laboratory of Regional Climate-Environment for Temperate East Asia, Institute of Atmospheric Physics, Chinese Academy of Sciences, Beijing 100029, China
[2] University of Chinese Academy of Sciences, Beijing 100049, China
[3] Key Laboratory of Middle Atmosphere and Global Environment Observation, Institute of Atmospheric Physics, Chinese Academy of Sciences,
Beijing100029, China
Correspondence to: Zhiwei Han ( hzw@mail.iap.ac.cn)

## Abstract

An online-coupled regional chemistry/aerosol-climate model (RIEMS-Chem) was developed and utilized to investigate the mechanisms of haze formation and evolution and aerosol radiative feedback during winter haze episodes in February-March 2014 over the Beijing-Tianjin-Hebei (BTH) region in China. Model comparison with a variety of observations demonstrated a good ability of RIEMS-Chem in reproducing meteorological variables, PBL heights, PM₂.₅ concentrations and its chemical components, as well as aerosol optical parameters. It was noteworthy that the model performances were remarkably improved for both meteorological variables and aerosol properties by taking aerosol radiative feedback into account, highlighting the necessity of developing online coupled chemistry-climate model. The weak southeasterly winds, high relative humidity and low PBL height favored accumulation and secondary formation of aerosols, resulting in a maximum daily and regional mean PM₂.₅ concentration exceeding 136 μg m⁻³ in the BTH region. The domain average aerosol radiative effects (AREs) were estimated to be -57 W m⁻² at the surface, 25 W m⁻² in the atmosphere and -32 W m⁻² at the top of atmosphere (TOA),



respectively, during the severe haze episode (20-26 February), and the maximum hourly ARE
at the surface reached -384 W m$^{-2}$ in the vicinity of Shijiazhuang in southern Hebei province
during this episode. The average feedback-induced changes in 2-m air temperature (T2),
10-m wind speed (WS10), 2-m relative humidity (RH2) and planetary boundary layer (PBL)
height over the BTH region during the haze episode were -1.8 °C, -0.5 m s$^{-1}$, 10.0% and -184
m, respectively. The domain average changes in PM$_{2.5}$ concentration due to the feedback
were estimated to be 20.0 μg m$^{-3}$ (29%) and 45.1 μg m$^{-3}$ (39%) for the entire period and the
severe haze episode, respectively, and they were enhanced to 21.1 μg m$^{-3}$ (36%) and 49.3 μg
m$^{-3}$ (49%) in terms of daytime mean during the haze episode, which demonstrated a
significant impact of aerosol radiative feedback on haze formation. The relative changes in
secondary aerosols were larger than those in primary aerosols, because chemical reactions
were also enhanced in addition to weakened diffusion by the feedback. The absolute change
in PM$_{2.5}$ concentrations caused by aerosol feedback was largest in the persistence stage,
followed by those in the growth stage and in the dissipating stage. Process analyses on haze
events in Beijing revealed that local emission, chemical reaction and regional transport
mainly contributed to haze formation in the growth stage, whereas vertical processes
(diffusion, advection and dry deposition) were major processes for PM$_{2.5}$ removals. Chemical
processes and local emissions dominated the increase in PM$_{2.5}$ concentrations during the
severe haze episode, whereas horizontal advection contributed to the PM$_{2.5}$ increase with a
similar magnitude to local emissions and chemical processes during a moderate haze episode
on 1-4 March. The contributions from physical and chemical processes to the
feedback-induced changes in PM$_{2.5}$ and its major components were explored and quantified
through process analyses. For the severe haze episode, the increase in the change rate of
PM$_{2.5}$ (9.5 μg m$^{-3}$ h$^{-1}$) induced by the feedback in the growth stage was attributed to the larger
contribution from chemical processes (7.3 μg m$^{-3}$ h$^{-1}$) than that from physical processes (2.2
μg m$^{-3}$ h$^{-1}$), whereas, during the moderate haze episode, the increase in the PM$_{2.5}$ change rate
(2.4 μg m$^{-3}$ h$^{-1}$) in the growth stage was contributed more significantly by physical processes
(1.4 μg m$^{-3}$ h$^{-1}$) than by chemical processes (1.0 μg m$^{-3}$ h$^{-1}$). In general, the aerosol-radiation
feedback increased the accumulation rate of aerosols in the growth stage through weakening
vertical diffusion, promoting chemical reactions, and/or enhancing horizontal advection. It



enhanced the removal rate through increasing vertical diffusion and vertical advection in the
dissipation stage, and had little effect on the change rate of PM$_{2.5}$ in the persistence stage.

## 1 Introduction

Aerosols affect radiation transfer by scattering or absorbing solar and infrared radiation,
by acting as cloud condensation nuclei (CCN) to modify cloud properties, and by heating the
atmosphere to alter cloud formation, termed as the aerosol direct radiative effect, indirect
effect, and semi-direct effect (Twomey, 1974; Albrecht, 1989; Ramanathan et al., 2001),
respectively. In addition, there exists a set of interactions between chemistry, radiation and
meteorology (Dawson et al., 2007; Zhang, 2008; Isaksen et al., 2009; Baklanov et al., 2014;
Cai et al., 2017), which is highly complex and nonlinear and is currently one of the least
understood mechanisms in air pollution and climate change. The above interactions are not
included or not well treated in current atmospheric models.
Rapid and continuous growth of economy and energy consumption in the past decades
has greatly elevated aerosol levels in China (Chan and Yao, 2008; Zhang et al., 2012; Li et al.,
2017a), resulting in serious air pollution problem and potentially significant influence on
radiation and climate at multi-scales. Although emission control strategies have been
gradually implemented in recent years, haze events still often occur in east China, especially
in north China in wintertime due to both higher anthropogenic emissions and poorer
meteorological conditions. The haze pollution issue has attracted wide attentions from public,
government and scientific community in China and a lot of monitoring and modeling studies
have been carried out to explore the sources, characteristics, formation and evolution
mechanisms of haze events at both urban and regional scales (Chan and Yao, 2008; Zhang et
al., 2012; Che et al., 2014; Guo et al., 2014; Huang et al., 2014; Sun et al., 2014; Zheng et al.,
2015; Cheng et al., 2016; Ding et al., 2016; Li and Han, 2016a; Cai et al., 2017; Fu and Chen,
2017; Li et al., 2017b; Wang et al., 2017; Zhang et al., 2018a; Zhong et al., 2018a; Zhong et
al., 2018b; An et al., 2019; Li et al., 2019a), through which our understanding on haze
pollution has been promoted. However, there is still a large gap in our knowledge about haze
formation mechanism, in particular the role of aerosol-radiation-meteorology feedback in the



formation and evolution of haze pollution.

The aerosol radiative feedbacks on air quality and meteorology have ever been studied in

American and Europe with regional online coupled meteorology-chemistry models, such as
WRF-Chem (Grell et al., 2005; Zhang et al., 2010; Forkel et al., 2012), which demonstrates
an important role of the feedback in both air quality and meteorology. Carslaw et al. (2010)
also pointed out the complexity and significance of natural aerosol interactions and feedbacks
within the Earth system.

In east Asia, Han et al. (2013) revealed a significant feedback of mineral dust   on dust

deflation and transport, atmospheric dynamics, cloud and precipitation in spring and an
improvement of model prediction for PM concentration and surface meteorology by the
inclusion of the feedback effect into an online coupled chemistry-aerosol-climate model. In
recent years, given the increasing concerns on haze pollution, some modeling studies have
been conducted to investigate the effect of aerosol radiative feedback on meteorology and
near surface $PM_{2.5}$ concentration, with focus on winter haze events in north China (Wang et
al., 2014a; Wang et al., 2014b; Zhang et al., 2015; Gao et al., 2015; Gao et al., 2016; Qiu et
al., 2017; Zhao et al., 2017; Zhang et al., 2018b; Chen et al., 2019; Wu et al., 2019). Most of
the model results exhibited a positive feedback which tended to increase $PM_{2.5}$ level, however,
the magnitude of such feedback differs largely, with the mean fractional change in $PM_{2.5}$
concentration varying from just a few percentage (Kajino et al., 2017; Wu et al., 2019) to
around 30% (Wang et al., 2014a). Some studies even show a negative feedback on $PM_{2.5}$ in
Beijing (Zhang et al., 2015; Gao et al., 2016). Recently, Gao et al. (2020) reported that the
aerosol–radiation feedback-induced daytime changes in $PM_{2.5}$ concentrations were less than 6%
during haze days in the BTH region in January 2010 from six applications of different online
coupled meteorology-chemistry models under the international framework of the MICS-Asia
(Model Inter Comparison Study for Asia) Phase III. There existed some differences in the
above modeling studies in terms of study period and haze pollution level, although they were
all for winter haze events in the BTH region. Zhong et al. (2018a) reported that over 70% of
$PM_{2.5}$ increase during cumulative explosive stage of haze event in Beijing in winter can be
attributed to the feedback effect based on integrated analysis of observations. The above
studies highlight the importance and large uncertainties in the aerosol radiative feedback,



which require further model development and investigation.
The diversity in the feedback effect among models could be associated with the
differences in the predictions of aerosol chemical components and aerosol optical properties,
the assumption of mixing state and hygroscopic growth scheme, as well as meteorological
fields, all of which determine the direction and magnitude of the feedback effect. Previous
model studies consistently underpredict PM concentrations, especially for aerosol
components, such as sulfate, nitrate and SOA concentrations, mainly due to incomplete
understanding and unrealistic treatment of secondary aerosol formation through multi-phase
chemical processes. Gao et al., (2018) reported that most of the participating models
(including WRF-Chem) in the MICS-Asia (Model Inter Comparison Study for Asia) project
underpredicted inorganic and organic aerosol concentrations by up to a factor of three.
Besides aerosol mass concentration, the unrealistic representation of aerosol properties, such
as composition, size distribution, mixing state, hygroscopic growth would also lead to model
biases in aerosol optical properties and direct radiative effects. The low biases in the
predicted aerosol compositions may lead to underpredictions of aerosol optical depth (AOD)
and consequently of aerosol radiative effects and feedback. Che et al. (2014) reported a
reduction of solar radiation by aerosols exceeding 200 W m$^{-2}$ during a severe haze event in
the north China Plain, much stronger than the estimations from models (around ~ -100 W
m$^{-2}$). Therefore, a realistic treatment and an accurate representation of aerosol processes and
properties are crucial to the estimation of aerosol radiative effects and feedback.
It has been well recognized that high aerosol loadings can apparently reduce incoming
solar radiation at the surface, leading to surface cooling and inversion associated with
reduced wind speed and vertical diffusivity, and consequently increase in surface aerosol
concentrations. However, while we have gained considerable knowledge on the overall
feedback effect of aerosols, the detailed processes involved in the feedback mechanism are
still poorly understood and barely quantified, for example, how does the aerosol radiative
effect modify meteorological variables? how do the radiative and meteorological changes
affect physical and chemical processes and in turn affect the magnitude and distribution of
aerosol components? How to quantify the relative contributions from various physical and
chemical processes to the feedback effect?



150 In this study, an online coupled regional climate-chemistry-aerosol model (RIEMS-Chem)

151 was developed and applied to explore the formation and evolution of haze pollution during

152 February-March 2014, in which a week-long haze episode with the daily maximum $PM_{2.5}$

153 concentration up to 400 $\mu g\ m^{-3}$ (hourly mean up to 483 $\mu g\ m^{-3}$) was observed. A wide variety

154 of field measurements of aerosol chemical components, optical properties, as well as

155 meteorological variables were conducted and applied to develop, constrain and validate the

156 model. The mechanisms of haze formation and evolution, aerosol radiative effects and

157 feedback on meteorology and chemistry were investigated and assessed. The overall aerosol

158 feedback on $PM_{2.5}$ and its aerosol compositions and the individual contributions to the

159 feedback from physical and chemical processes (advection, diffusion, deposition, chemistry,

160 etc.) during haze events were interpreted and quantified by a process analysis approach

161 incorporated in the model. The results from this study is expected to provide new insights

162 into the mechanism of aerosol-radiation-meteorology feedback, which is currently the source

163 of one of the largest uncertainties in haze pollution formation and evolution.

## 2 Model and Data

166 2.1 Model description

167 An online-coupled regional atmospheric chemistry/aerosol-climate model RIEMS-Chem

168 was used in this study, which was developed based on the Regional Integrated Environmental

169 Model System (RIEMS). A series of modules and parameterizations were adopted to

170 represent major physical processes, including a modified Biosphere-Atmosphere Transfer

171 Scheme (BATS; Dickinson et al., 1993) to simulate land surface process, the Medium-Range

172 Forecasts scheme (MRF) to represent the planetary boundary layer process (Hong and Pan,

173 1996), the cumulus convective parameterization scheme from Grell (1993), and a modified

174 radiation package of the NCAR Community Climate Model, version CCM3 (Kiehl et al.,

175 1996) to represent radiation transfer process including aerosol effect. RIEMS had been

176 applied to investigate East Asian monsoon climate and the interactions among physical,

177 biological and chemical processes (Xiong et al., 2009; Zhao, 2013; Wang et al., 2015).

178 RIEMS had participated in the Regional Climate Model Intercomparison Project (RMIP) for



Asia and it was one of the best models in predicting air temperature and precipitation over
east Asia (Fu et al., 2005).
The online-coupled model RIEMS-Chem has been developed in recent years by
incorporating major atmospheric chemistry/aerosol processes into the host model. Pollutants
are driven by meteorological fields provided by RIEMS and feedback to the existing dynamic
and physical modules (Han, 2010; Han et al., 2012). Major atmospheric processes including
emission, advection, diffusion, multi-phase chemistries, dry deposition and wet scavenging of
pollutants are considered. The advection and diffusion for pollutants are treated with the same
scheme for substances (such as moisture). Gas phase chemistry is represented by an updated
Carbon-bond mechanism (CB-IV; Gery et al., 1989). Thermodynamic processes are
calculated by the ISORROPIA II model (Fountoukis and Nenes, 2007). Dry deposition
velocity of aerosol is calculated by a size-dependent scheme which is expressed as the inverse
of the sum of resistance plus a gravitational settling term, while below-cloud wet scavenging
of aerosol is parameterized as a function of precipitation rate and collision efficiency of
particle by hydrometeor (Han et al., 2004). Heterogeneous reactions between gases and
mineral dust and sea salt aerosols have also been incorporated into RIEMS-Chem (Li and
Han, 2010; Li et al., 2018a). SOA formation is parameterized by a two-product model (Odum
et al., 1997).
Current atmospheric chemistry models generally tend to underpredict sulfate
concentrations, especially in source regions during wintertime, such as north China, which
could be due to uncertainties in the treatment of chemical formation mechanism. Recent
model studies suggested that heterogeneous reactions could be an important pathway in
sulfate formation during winter haze episodes in north China (Li et al., 2017c; Li et al.,
2018b). Therefore, heterogeneous reactions concerning the conversion of $SO_2$ to sulfate on
pre-existed hydrated aerosols were incorporated in RIEMS-Chem. The method of Li et al.
(2018b) was adopted, in which the uptake coefficient ($\gamma_{so2}$) was a stepwise function
determined by the aerosol water content (awc) which was predicted by the ISORROPIA II
model. Accordingly, the upper bound of awc was set to 300 $\mu g\ m^{-3}$ ($\gamma_{so2}=1\times10^{-4}$) while the
lower bound was 30 $\mu g\ m^{-3}$ ($\gamma_{so2}=1\times10^{-6}$). $\gamma_{so2}$ was linearly interpolated between the upper
and lower bounds in terms of awc.



RIEMS-Chem treats 9 aerosol types including sulfate, nitrate, ammonium, black carbon
(BC), primary organic aerosol (POA), secondary organic aerosol (SOA), anthropogenic
primary PMs ($PM_{2.5}$ and $PM_{10}$), dust and sea salt. The size distribution of the different types
of aerosols is previously prescribed based on the OPAC database (Optical Properties of
Aerosols and Clouds) (Hess et al., 1998). In this study, measurements in Beijing are used to
represent aerosol size distribution more realistically and to constrain the model. During the
study period, a scanning mobility particle sizer (SMPS; TSI, Inc., Shoreview, MN, USA) was
used to measure aerosol size distribution (Ma et al., 2017) and the geometric mean radius of
inorganic, black carbon and organic carbon aerosols were estimated to be 0.1 μm, 0.05 μm
and 0.1 μm, with standard deviations of 1.65, 1.6, 1.65, respectively. The above aerosol size
information was incorporated into RIEMS-Chem. The deflation of mineral dust is represented
by the scheme of Han et al. (2004) with 5 size bins (0.1–1.0, 1.0–2.0, 2.0–4.0, 4.0–8.0, 8.0–
20.0μm). Primary PMs from anthropogenic are also assigned to the 5 size bins.
Recent observational analyses of aerosol mixing state in Beijing (Ma et al., 2012; Wu et
al., 2016) indicated that more than 80% aerosols were internally mixed with BC during haze
days, whereas about 70% of aerosols were externally mixed with BC in clean days, so an
internal mixing assumption was adopted in this study because the focus is paid on haze
episode.
A κ-Köhler theory (Petters and Kreidenweis, 2007) was used to parameterize aerosol
hygroscopic growth. In the model, the κ values for inorganic aerosols, BC, POA, SOA,
dust/primary PMs and sea salt were set to 0.65, 0, 0.1, 0.2, 0.01 and 0.98, respectively,
according to previous observational and modeling studies (Riemer et al., 2010; Liu et al.,
2010a; Westervelt et al., 2012).
The refractive index of an internally mixed aerosol is calculated by volume-weighting
the refractive indices of each aerosol component and water. Aerosol optical parameters
including extinction coefficient, single scattering albedo and asymmetry factor were
calculated by a Mie-theory based method developed by Ghan and Zaveri (2007), in which
aerosol optical parameters were pre-calculated with the Mie code and then fitted by
Chebyshev polynomials to create a lookup table of polynomial coefficients. By this way, the
calculation of aerosol optical parameters was much faster than using the traditional Mie




239 theory solution with a similar level of accuracy. This method has been successfully used in

240 the estimation of AOD over East Asia (Han et al., 2011a).

241  RIEMS-Chem has been successfully applied in previous modeling studies of

242 anthropogenic aerosols, mineral dust and marine aerosols regarding spatial-temporal

243 distributions, physical and chemical evolutions, radiative and climatic effects over east Asia

244 (Han et al., 2011b; 2012; 2013; 2019; Li et al., 2014; Li and Han, 2016b; 2016c; Li et al.,

245 2019b). RIEMS-Chem have been participating in the international model comparison project

246 Model Inter Comparison Study for Asia phase III (MICS-Asia III) and shows a good ability

247 in predicting $PM_{2.5}$ concentration and AOD over East Asia (Gao et al., 2018).

249 2.2 Process analysis

250  In RIEMS-Chem, a time-splitting scheme based on continuity equation is applied to

251 predict species concentrations; therefore, the species concentrations are the net results of

252 successive changes in concentration due to different atmospheric physical and chemical

253 processes, and the changes in species concentration by each process can be recorded,

254 allowing the quantification of individual contribution of each process to species variation. In

255 this study, a process analysis (PA) scheme, which calculates the Integrated Process Rates

256 (IPR) at each time step and each grid, was embedded in RIEMS-Chem to identify the

257 contributions of physical and chemical processes to aerosol evolution. At each time step, the

258 IPR for a certain process was calculated by subtracting the species concentrations at the

259 beginning of this process from the ones after the process. The IPR method has ever been

260 applied to study the formation and fate of particulate and gaseous pollutants in North America

261 and China (e.g. Yu et al., 2008; Zhang et al., 2009; Liu et al., 2010b). The processes involved

262 in aerosol evolution include emissions of primary species, advections (horizontal and

263 vertical), diffusions (horizontal and vertical), dry deposition, chemical processes (gas-phase

264 chemistry, thermodynamic equilibrium and heterogeneous reactions), cloud processes and

265 wet deposition. Here cloud process represents the effects of cloud attenuation of photolysis

266 rate, aqueous-phase chemistry and in-cloud mixing. In this study, PA is applied not only to

267 quantify the contributions of individual physical and chemical processes to haze evolution,

268 but also to help interpret the processes involved in aerosol radiative feedback. In addition,



different from the previous PA application, chemical processes are further classified into gas
phase, thermodynamic and heterogeneous reactions to provide more details on chemical
pathways of secondary aerosol formation. The mass balance of IPR has been examined,
assuring that the change in species concentration during one time step is equal to the sum of
IPRs by each of the processes.

2.3 Emission inventories

Monthly mean anthropogenic emissions of sulfur dioxide ($SO_2$), nitrogen ($NO_x$),

ammonia ($NH_3$), non-methane volatile organic compounds (NMVOC), carbon monoxide
(CO), black carbon (BC), primary organic carbon (POA), other anthropogenic primary $PM_{2.5}$
and primary $PM_{10}$ in China for the year 2014 were obtained from the MEIC inventory
(Multi-resolution Emission Inventory for China) which was developed by Tsinghua
University (http://meicmodel.org). Anthropogenic emissions outside China were taken from
the MIX inventory which was developed to support the Model Inter-Comparison Study for
Asia phase III (MICS-Asia III) and the Hemispheric Transport of Air Pollution (HTAP)
projects (Li et al., 2017a). Both inventories of MEIC and MIX have the horizontal resolution
of 0.25 degree. Biomass burning emissions of aerosols and gas precursors for the year 2014
with a horizontal resolution of 0.25 degree were derived from the fourth version of the Global
Fire Emissions Database (GFED4) (Giglio et al., 2013). Monthly mean biogenic emissions of
isoprene and monoterpene were derived from Global Emissions Inventory Activity (GEIA,
http://www.geiacenter.org/). The above emission data were bilinearly interpolated to the
lambert projection of RIEMS-Chem.

2.4 Model configuration and numerical experiments

RIEMS-Chem was configured on a lambert conformal projection with horizontal

resolution of 60 km, covering most areas of China, the Korean Peninsula, Japan and part of
the Indo-China Peninsula (Figure 1). 16 vertical layers distributed vertically and unevenly in
the terrain-following sigma coordinate, with the lowest 8 layers within the boundary layer.
This study focused on the Beijing-Tianjin-Hebei (BTH) region with more attentions to the
Beijing metropolitan. The study period was from 10 February to 12 March, 2014,





encountering several haze episodes. The first 7 days were taken as model spin-up and the
results from 17 February to 12 March were used for analysis.
Initial and boundary conditions for meteorological variables were provided by the final
reanalysis data (FNL) with 1°×1° resolution and 6-hourly interval from the National Centers
for Environmental Prediction (NOAA/NCEP, 2000). Lateral boundary conditions of chemical
species at 6-hourly interval were derived from the simulations of the global chemical model
MOZART-4 (Model for Ozone and Related chemical Tracers, version 4; Emmons et al.,

2010).

To investigate the aerosol radiative effects and its potential feedback on solar radiation,
meteorological variables, planetary boundary layer (PBL) and aerosol concentrations in the
study domain, two simulations were designed. The FULL simulation (with aerosols)
considered all aerosol direct and indirect effects and feedbacks; the NoAer simulation
(without aerosols) shut off all aerosol radiative effects. In both simulations, the driving
meteorological data, emissions and model settings were exactly the same.

2.5 Observational data
Several observational datasets for meteorological variables, aerosol concentrations and
aerosol optical parameters were obtained and used for model comparison and analysis.
In-situ 3-hourly observations of temperature at 2 meter (T2), wind speed at 10 meter
(WS10) and relative humidity at 2 meter (RH2) from three meteorological monitoring sites
around Beijing (Figure 1) were collected from the China Meteorological Data Service Center
(CMDS) (http://data.cma.cn/).
To evaluate the model ability in reproducing evolution of planetary boundary layer
(PBL), high-frequency sounding data measured around 14:00 LST at the Xianghe station
(39°45′N, 116°58′E; approximately 63 km southeast of Beijing downtown) were collected,
from which the PBL height can be determined based on the vertical gradients of virtual
potential temperature and water mixing ratio according to the method from Heo et al. (2003).
This sounding dataset provided a good indicator of mixing layer height because the sounding
was launched at 14:00 LST and lasts for about one hour. The meteorological sounding was
launched in Xianghe once a week (every Tuesday) and totally four soundings were available





during the study period (18 and 25 February, 4 and 11 March). Fortunately, the four
soundings encountered one severe haze episode, one moderate haze episode, and two clean
days, providing robust evidences on day-to-day variation of mixing layer height under
various atmospheric conditions. Hourly downward shortwave radiation flux (SWDOWN) at
the surface was measured simultaneously at the Xianghe station by a pyranometer with sun
shield and was used in this study.

The measurements of mass concentrations of $PM_{2.5}$ and its components and aerosol

optical parameters were carried out at the tower division of the Institute of Atmospheric
Physics (IAP), Chinese Academy of Sciences (CAS) in Beijing (39°58′N, 116°22′E) from 17
February to 12 March, 2014. Real-time hourly $PM_{2.5}$ mass concentrations were online
measured by a hybrid beta attenuation particulate monitor (Model 5030 SHARP, Thermo
Scientific, USA). $PM_{2.5}$ samples were collected in parallel by an R&P Partisol®Model 2025
dichotomous sequential PM air sampler (Thermo, USA) and a MiniVol TAS PM sampler
(Airmetrics, USA) between 24 February and 12 March, 2014. Samples were collected twice
per day with one during the daytime (from 7:00 to 19:00 LST) and the other at night (from
19:00 to 7:00 of the next day). Totally 33 half-day samples were collected. Aerosol chemical
compositions including sulfate ($SO_4^{2-}$), nitrate ($NO_3^-$), ammonium ($NH_4^+$), BC and OC were
analyzed by ion chromatography (Dionex ICS-90 for cations and ICS-1500 for anions) and a
DRI-2100A carbonaceous aerosol analyzer. Real-time hourly aerosol extinction coefficient
and aerosol absorption coefficient at dry condition (RH=10%) were synchronously measured
by a nephelometer (Aurora3000) and an aethalometer (AE-31), respectively. Detailed
information about this experiment including the sampling site, instruments, measurement
procedures and sample analysis were well documented in Ma et al. (2017). The mass
concentration of SOC was estimated using a revised EC tracer method (Zhao et al., 2013).

Measurements of AOD at the 4 sites (Nanjiao, Tianjin, Gucheng and Shangdianzi) in the

BTH region were obtained from the China Aerosol Remote Sensing Network (CARSNET)
(Che et al., 2014). Nanjiao is an urban site located in southern Beijing. Tianjin site is located
in the center of Tianjin city, about 120 km to the southeast of Beijing. Gucheng, a suburban
site in Hebei province, is about 130 km to the southwest of Beijing downtown. Shangdianzi is
located 150 km to the northeast of Beijing, which is a background station since it is far away



359 from anthropogenic sources. Daily mean AOD was derived by temporally averaging the raw

360 data measured by sunphotometer during daytime. To compare with the model output, AOD at

361 550 nm was used.


363 **3 Model validations**

364 3.1 Meteorological variables

365  Wind speed, temperature and relative humidity are key meteorological factors affecting

366 physical and chemical processes of atmospheric pollutants. The statistics for comparison

367 between in-situ observation and the FULL simulation for WS10, T2 and RH2 are presented in

368 Table 1. At the 3 sites (Beijing, Tianjin and Tanggu), the model performances were

369 reasonably good, although wind speeds were somewhat overpredicted. The overall

370 correlation coefficient (R) and normalized mean bias (NMB) at the 3 sites were 0.83 and -2%

371 for T2, 0.61 and -1% for RH2 and 0.47 and 31% for WS10. In all, RIEMS-Chem was able to

372 reasonably reproduce the meteorological variables during the study period. The statistics for

373 NoAer simulation are also list in Table 1. It is noteworthy that the statistics for the FULL

374 simulation are overall better than those for NoAer simulation, such as the warm bias in the

375 simulated air temperature and positive bias in wind speed are apparently reduced. This

376 demonstrates the inclusion of aerosol radiative effects does improve meteorological

377 prediction in this study.

378  The observed hourly SWDOWN in Xianghe was compared with model simulation

379 (Figure 2a). In general, the FULL case well reproduced SWDOWN in clean days and

380 light-moderate polluted days, but tended to underpredict observations in heavy haze days,

381 such as the period from 20 to 26 February. Underpredictions of cloud amount and PM

382 concentrations could be reasons for the low bias. For the entire study period, the observed

383 and simulated (FULL) mean SWDOWN were 136.0 W/m$^2$ and 188.4 W/m$^2$, respectively,

384 with R of 0.91 (Figure 2a). If only days with low cloud covers were considered, the

385 SWDOWNs were 183.3 W/m$^2$ and 213.7 W/m$^2$ from observation and the FULL case,

386 respectively, with the NMB of 16%. In contrast, the NoAer case failed to capture the

387 decreasing tendency of SWDOWN during haze days, resulting in a larger bias (NMB of 72%)



than the FULL case.

3.2 Planetary boundary layer (PBL) height
Figure 2b shows the simulated PBL heights at 14:00 LST from the FULL case and
NoAer case during the study period and the observed PBL heights at 14:00 LST determined
from air soundings on 18 February (clean), 25 February (severe haze), 4 March (clean) and
11 March (haze), 2014, respectively. There was large variation in PBL height in the afternoon,
with higher PBL height in clean days and lower one in haze days, inversely related to the
$PM_{2.5}$ level. The FULL case well reproduced the very low PBL height during the most severe
haze episode on 25 February, with the observed and simulated PBL heights to be 569m and
587m, respectively. In clean days, the much higher mixing layer was also well captured, such
as, on 4 March, the observed and simulated PBL heights were 2305m and 2535m,
respectively. It is noteworthy that the simulated PBL heights in the NoAer case were
consistently higher than those in the FULL case, and the PBL height simulation from the
FULL case (considering aerosol radiative effects) was apparently in a better agreement with
observation than that from the NoAer case, except for that on 18 February.

3.3 Mass concentrations of $PM_{2.5}$ and aerosol components
Figure 2c shows the hourly $PM_{2.5}$ mass concentrations observed at the IAP site and those
from the FULL simulation and NoAer simulation. The study period was characterized by
three haze episodes, which was the episode 1 on 20-26 February, the episode 2 from 1 to 4
March, and the episode 3 from 8 to 11 March. The first episode experienced the most severe
pollution with the maximum hourly $PM_{2.5}$ concentration exceeding 480 μg m$^{-3}$ on 25
February. The second and third ones were moderately polluted in terms of magnitude and
lasting time. In general, the model reproduced the hourly variation of $PM_{2.5}$ concentrations
reasonably well in the FULL case, although the peaks were somewhat underpredicted in
some days, which could be partly due to the overprediction of wind speed (Table 1) and
potential uncertainties in emission inventories. The low bias in $PM_{2.5}$ concentrations could
also contribute to the overprediction of SWDOWN during the first haze episode (20-26
February) discussed in section 3.1. The average $PM_{2.5}$ concentrations during the study period


were 142.0 μg m$^{-3}$ and 131.4 μg m$^{-3}$ from observation and the FULL simulation, respectively,
with R of 0.8 and NMB of -7% (Table 2), which demonstrates a good model performance for
PM$_{2.5}$ predictions for the winter haze periods. A remarkable feature shown in Figure 2 is the
significant negative correlation between PM$_{2.5}$ concentration and PBL height and SWDOWN.
The comparison between the simulated daily mean surface aerosol components (sulfate
(SO$_4^{2-}$), nitrate (NO$_3^-$), ammonium (NH$_4^+$), BC and OC) and observations at the IAP site are
presented in Figure 3. The daily mean observation in the figure is an average of the half-day
samples, while the original half-day samples are used for statistics calculation in Table 2. The
model (from the FULL case) generally exhibits a good performance for inorganic aerosol
(sulfate, nitrate and ammonium) concentrations in terms of both daily variation and
magnitude (Figure 3a - 3c). It is encouraging that the maximum values on 25 February during
the first haze episode and the moderate values on 3 March in the second haze episode are
well reproduced, although some low biases occurred in the last few days. On average, the
model simulations of 20.3 μg m$^{-3}$, 24.3 μg m$^{-3}$ and 13.9 μg m$^{-3}$ are very close to the
observations of 21.0 μg m$^{-3}$, 26.0 μg m$^{-3}$ and 14.1 μg m$^{-3}$ for sulfate, nitrate and ammonium,
respectively, with Rs of 0.92, 0.88 and 0.91 and NMBs of -4%, -6% and -2%, respectively
(Table 2). Most of the online coupled models tended to underpredicted sulfate concentration
(Gao et al., 2016; Qiu et al., 2017; Gao et al., 2018), which led to an underestimation of
aerosol optical depth and radiative effect. The model in this study improves the simulation of
inorganic aerosols, mainly through the inclusion of heterogeneous chemical reactions for
inorganic aerosols.
The model also reproduced the temporal variation and magnitude of BC (Figure 3d) and
OC (Figure 3e) concentrations in Beijing reasonably well. However, the model tended to
underpredict the peak OC values on 24-25 February and to overpredict BC concentrations
from late February to early March. The low bias in OC simulation during the haze episodes
could be attributed to the underprediction of SOC (Figure 3f) due to potentially missing
chemical pathways. Uncertainties in the emission inventory could also be a reason. Li et al.,
(2017a) reported the uncertainties in BC and OC emissions for China could be ±200%, larger
than those of emissions for gases (<70%) and primary PMs (~130%). The period mean BC
concentrations from observation and simulation were 5.2 μg m$^{-3}$ and 6.7 μg m$^{-3}$, respectively,



with R of 0.92 and NMB of 28% (Table 2). The period mean simulated and observed POC
concentrations were 18.4 $\mu g\ m^{-3}$ and 15.5 $\mu g\ m^{-3}$, respectively, with R of 0.93, whereas the
simulated SOC concentration was 9.9 $\mu g\ m^{-3}$, lower than observation (13.6 $\mu g\ m^{-3}$) by 27%,
with a correlation coefficient of 0.56. For OC (sum of POC and SOC), the simulated value
(28.3 $\mu g\ m^{-3}$) was very close to the observation (29.1 $\mu g\ m^{-3}$), with R of 0.88 and NMB of
-3%, respectively, which indicated a generally good model performance for the total OC
concentration.
It is noteworthy that by considering aerosol radiative effects, the model apparently
improved simulations for both $PM_{2.5}$ and its chemical compositions, which is illustrated by
comparing model results between the FULL and NoAer cases (Figure 2c, Figure 3 and Table
2). Another important finding is that the duration of haze episode was prolonged by about 2-3
hours by the aerosol radiative feedback compared with that without aerosol feedback (Figure
2c).

3.4 Aerosol optical parameters
Figure 4a and 4b show the measured and simulated hourly aerosol extinction coefficient
(EXT) and aerosol absorption coefficient (ABS) at an RH of 10% at the IAP site during the
study period. It clearly showed that the model was able to well reproduce the magnitudes and
temporal variations of EXT and ABS under dry condition in the FULL case, although the
model tended to predict higher ABS in some days possibly due to the overprediction of BC
concentration. Single scattering albedo (SSA) which is defined as the ratio of scattering
coefficient (EXT minus ABS) to extinction coefficient is also given in Figure 4c. The FULL
case generally simulated high SSA values during haze episodes, such as 0.92 on 20-26
February, 0.85-0.9 from 1 to 4 March and 0.8-0.9 on 8-11 March, suggesting a dominant role
of light scattering aerosols in haze days. It is encouraging that the model reproduced SSA
during the severe haze episode (on 20-26 February) quite well, with both the simulation and
observation being approximately 0.92. However, SSA observation in clean days (such as on
5-7 March) was lower than that in haze days, and the model tended to overpredict SSA in
clean days, which could be attributed to uncertainties in measurement. In clean days, both the
denominator (EXT) and numerator (EXT minus ABS) were small, a subtle perturbation in





EXT and/or ABS can result in a large variation in SSA. A previous observational study in
Beijing suggested that SSA observation was more uncertain in clean days than in polluted
days because the observed aerosol extinction coefficient was too low in clean days (Jing et al.,
2015). On average, the observed EXT, ABS and SSA values were 0.51 km$^{-1}$, 0.048 km$^{-1}$ and
0.85, respectively, whereas, the corresponding FULL simulations were 0.53 km$^{-1}$, 0.052 km$^{-1}$
and 0.88, with Rs of 0.8, 0.7 and 0.7 and NMBs of 4%, 10% and 5%, respectively (Table 2).
The above comparison demonstrates a good ability of the model in estimating aerosol optical
properties during the study period, which could be attributed to both the good performance
for aerosol compositions and the realistic representation of aerosol properties (aerosol size
distribution, mixing state, hygroscopic growth etc.), which is based on real-time
measurements in Beijing.

Besides EXT and ABS measured under dry condition, measurements of AOD at the four

CARSNET sites around Beijing (Nanjiao, Tianjin, Gucheng and Shangdianzi) were also used
to evaluate the model ability in simulating aerosol optical parameters in real atmosphere
(Figure 5). At the Nanjiao site, which is about 50km southeast of Beijing downtown (Figure
5a), AOD measurement was unavailable in most days during the first haze episode (20 to 26
February), with only two data (around 4.8) available on 25 February. The simulated daily
AOD from the FULL case varied from 3.1 to 4.0 during 24 - 26 February, somewhat lower
than the observation. The model tended to simulate lower AOD during the third haze episode
(8 to 11 March), which can be partly attributed to the predicted lower aerosol concentrations.
The measured AODs in Gucheng (southwest to Beijing) and Tianjin were similar in terms of
variation and magnitude (Figures 5b and 5c), showing high values during pollution periods
with the maximum daily AOD exceeding 4.0 in Gucheng and 3.5 in Tianjin. The FULL case
reproduced the AOD variations and magnitudes reasonably well at the two sites although low
biases still occurred during 8 to 11 March in Gucheng. For the regional background site
Shangdianzi (Figure 5d), the magnitude and variation of AOD were similar to those in
Nanjiao, suggesting that the haze episodes were regionally distributed because the temporal
variations and magnitudes of AOD were generally consistent at the four sites.

Table 3 summaries the performance statistics for daily mean AOD. In general, the model

reproduced the temporal variation and magnitude of AOD around Beijing reasonably well





with the overall R of 0.81 (0.67 ~ 0.90) and NMB of -8.6% (-15.6% ~ 6.2%). The
underestimation is mainly contributed by the low biases during the third haze episode (8 to 11
March) when inorganic aerosol concentrations were underestimated (Figure 3a-3c). In
addition, the limitation in AOD samples during the severe haze episode in Nanjiao and
Shangdianzi could amplify the negative bias. At the Gucheng and Tianjin sites where more
samples were available, the mean measured AODs were 1.7 and 1.4, respectively, agreeing
well with the simulated values of 1.5 and 1.3 from the FULL case.

In summary, the above comparisons demonstrate that RIEMS-Chem was capable in

reproducing the spatial distribution and temporal variation of meteorological variables (air
temperature, wind speed, surface shortwave radiation, PBL height etc.), concentrations of
total $PM_{2.5}$ mass and its chemical compositions and aerosol optical properties during the
winter haze periods around Beijing. It is also noteworthy that the inclusion of aerosol
radiative effects apparently improved the overall model performance for both meteorological
variables and aerosol physical and chemical properties, highlighting the necessity to develop
online coupled chemistry-meteorology model for both air quality and climate research. The
good agreement above increases confidence in the reliability of the following model results
on aerosol radiative effects and feedback.

**4 Model results**
4.1 Distributions of meteorological variables and near surface $PM_{2.5}$ concentration

The period-mean distributions of near-surface wind speed (WS10), temperature (T2),

relative humidity (RH2), PBL height and $PM_{2.5}$ concentration are shown in Figures 6a to 6e.
During the study period, westerly winds dominated the northwestern parts of the BTH region
while southeasterly prevailed over the southeastern parts, as a result, the near-surface wind
speeds were fairly weak over the convergence zone from southern Hebei province to Beijing
(Figure 6c). Such wind pattern indicated that pollutants from southern parts of the domain
(such as Shandong and Henan provinces) can be transported northward to Beijing, Tianjin
and Hebei, and air pollutants over the weak-wind regions were easily accumulated to high
level. Near-surface temperature showed an apparent south-to-north gradient, with surface air



temperature in a range of 4 °C to 6 °C over the southern BTH region, -2 °C to 2 °C in the vicinity of Beijing and parts of central Hebei, and lower than -2 °C in northern parts of the domain (Figure 6a). Relative humidity was higher (~65% to 75%) over northern areas and lower (~55% to 65%) over southern areas (Figure 6b). PBL height also exhibited an apparent gradient in spatial distribution (Figure 6d), ranging from 800-1000 m in northern Hebei and Inner Mongolia to about 600-700 m in southern Beijing, Tianjin and southern Hebei. A belt of high $PM_{2.5}$ concentration spread from southwest to northeast (Figure 6e), with the maximum value up to 150 $\mu g \ m^{-3}$ in the vicinity of Shijiazhuang and Beijing and Tianjin. The regions with high $PM_{2.5}$ concentrations generally corresponded well to the weak-wind areas shown in Figure 6c.

Averaged over the BTH region and the entire study period, the simulated T2, WS10, RH2, PBL height and $PM_{2.5}$ concentration from the FULL case were 0.61 °C, 1.2 $m \ s^{-1}$, 67.0%, 698.4 m and 90.0 $\mu g \ m^{-3}$, respectively. According to the "Technical Regulation on Ambient Air Quality Index" prescribed by Chinese Ministry of Environmental Protection in 2012, a pollution event occurs when 24-hr mean $PM_{2.5}$ concentration $\geq$ 75 $\mu g \ m^{-3}$. Totally, there were 11 days with domain and daily average $PM_{2.5}$ concentration exceeding 75 $\mu g \ m^{-3}$ in the BTH region, with the maximum exceeding 136 $\mu g \ m^{-3}$, indicating the severity of air pollution during the study period.

4.2 Distributions of AOD, SSA and aerosol direct radiative effects

Figure 6f shows that high AODs mainly distributed from northern Beijing to southwestern Hebei, southern Shanxi and northern Henan provinces, with the maximum up to 1.1. As AOD was determined by vertical profiles of aerosol compositions and RH, the spatial distribution of AOD was somewhat different from that of $PM_{2.5}$ concentration. During the study period, the regional mean AOD in the BTH region was 0.78 (Table 4), about twice the long-term observed value of about 0.4 in February and March in the same region (Song et al., 2018).

The simulated SSAs were above 0.88 in the BTH region (Figure 6g), with relatively lower values (0.88 - 0.9) in the areas of high $PM_{2.5}$ concentration and higher ones (0.92 - 0.98) in the relatively clean areas. On average, the simulated SSA in the BTH was 0.91 (Table 4),





within the range of 0.87 to 0.95 measured in the same region in January 2013 (Che et al.,
2014) but slightly lower than the model simulated annual mean of 0.95 over eastern China
(Zhuang et al., 2013).
All-sky aerosol radiative effects at the surface ($ARE_{surf}$), at the top of atmosphere
($ARE_{TOA}$) and in the atmosphere ($ARE_{atm}$) under all-sky condition are presented in Figures 6h
to 6j. During the study period, aerosols induced a negative ARE both at the surface and TOA
and a positive ARE in the atmosphere over the BTH. The distribution of ARE resembles that
of AOD, generally showing stronger effects over southwestern Hebei, Shanxi and northern
Henan provinces where high AOD occurred. Moderate AREs appeared over Beijing, Tianjin
and central Hebei, while relatively weak AREs appeared over the northern domain. The
domain average AREs in the BTH region during the period were estimated to be -37 W m$^{-2}$,
19 W m$^{-2}$ and -18 W m$^{-2}$ at the surface, in the atmosphere and at the TOA, respectively (Table
4). The indirect radiative effect was also estimated to be about -2 W m$^{2}$ at the surface and the
TOA on average, much smaller than the direct radiative effect; therefore, the total radiative
feedback is predominated by direct radiative effect during the study period.
The domain average all-sky AREs during the first haze episode (20-26 February) were
-57 W m$^{-2}$, 25 W m$^{-2}$ and -32 W m$^{-2}$ at the surface, in the atmosphere and at the TOA,
respectively, and the values were further enhanced to -123 W m$^{-2}$, 53 W m$^{-2}$ and -70 W m$^{-2}$ in
terms of daytime mean. The maximum AREs at the surface and at TOA reached -384 W m$^{-2}$
and -231 W m$^{-2}$, respectively, at 13:00 LST on 23 February in the vicinity of Shijiazhuang.
In Beijing, the estimated mean AREs were -70 W m$^{-2}$, 32 W m$^{-2}$ and -38 W m$^{-2}$ at the
surface, in the atmosphere and at the TOA, respectively, during the first haze episode,
whereas the maximum ARE at the surface reached -304 W m$^{-2}$ at 13:00 LST on 22 February,
which was associated with the high $PM_{2.5}$ concentration (453 μg m$^{-3}$) at that time.
Based on in-situ surface measurements, Che et al. (2014) estimated that during haze
periods in January 2013, the mean daytime AREs at Nanjiao and Xianghe were
approximately -42 W m$^{-2}$ and -50 W m$^{-2}$ at TOA, and -120 W m$^{-2}$ at the surface at both sites.
In this study, the daytime AREs averaged over the severe haze period (20-26 February) at
TOA were estimated to be -77 W m$^{-2}$ and -74 W m$^{-2}$ at Nanjiao and Xianghe, while the
corresponding AREs at the surface were -146 W m$^{-2}$ and -140 W m$^{-2}$, respectively. Che et al.



(2014) also reported the maximum daily mean surface ARE of -220 W m$^{-2}$ at Nanjiao during a severe haze episode in January 2013, in this study, the corresponding ARE was estimated to be approximately -200 W m$^{-2}$ at the same site during the severe haze episode in February 2014. Therefore, the magnitudes of AREs during haze episodes simulated from this study agreed favorably with the above observational based estimations around Beijing, despite the different time period.

4.3 Impacts of aerosol radiative feedback on meteorological variables and aerosols

Figure 7a-7e shows the mean differences in T2, RH2, wind speed, PBL height and near surface PM$_{2.5}$ concentration induced by the radiative feedback due to all aerosols (FULL minus NoAer) in the domain during the study period.

The aerosol radiative effects led to a reduction in surface shortwave radiation and thus surface air temperature in the entire domain. The magnitude of T2 variation decreased from south to north of the BTH, with -1.6 °C to -2 °C in southern Hebei and -1.2 °C to -1.8 °C in southern Beijing, respectively. Correspondingly, RH2 increased by 10%-16% in the above regions. The changes in wind speed showed a patchy pattern, with decreases by ~0.1 m s$^{-1}$ in southern Hebei, increases by ~0.2 m s$^{-1}$ in central Hebei, and decreases in most parts of Beijing. Wind vector shows an anomalous northerly wind of ~0.5 m s$^{-1}$ in the BTH region. Due to the reduction in surface shortwave radiation, PBL height decreased over the entire region, with the maximums up to 240 m in southern Hebei and northern Tianjin. The changes in PBL height varied from -210 m in southern Beijing to -90m in northern Beijing. PM$_{2.5}$ concentrations were consistently enhanced over the entire region, with the maximum increase up to 33 μg m$^{-3}$ in southern Hebei and portions of Beijing and Tianjin. In most of the BTH region, the percentage increase of PM$_{2.5}$ exceeded 25%, with the maximum increase exceeding 33% in the vicinity of Shijiazhuang. It is of interest that the regions with the maximum increase of PM$_{2.5}$ generally corresponded to those with the maximum decrease in PBL height. The presence of aerosols reduced solar radiation reaching the ground surface, resulting in decreases in surface air temperature and PBL height and an increase in relative humidity, all of which favored accumulation and formation of aerosols due to weakened vertical mixing and enhanced secondary aerosol formation.



The aerosol feedback during the first haze episode was further explored due to the much
higher PM$_{2.5}$ level than the period average. Figure 7f-7j show the mean changes in
meteorological variables and PM$_{2.5}$ concentrations during the first haze episode (20-26
February). In general, the changes induced by aerosol feedback were larger during the severe
haze episode than those over the entire study period. T2 decreased by 1.8 °C to 2.7 °C along
with an increase up to 20% in RH in southern Hebei and southern parts of Beijing and Tianjin.
Different from the entire period average, wind speed decreased consistently in the BTH, with
a maximum decrease of 1 m s$^{-1}$. PBL height decreased by ~300 m in southern Hebei,
corresponding to the areas with large air temperature decrease. This resulted in a consistent
increase in PM$_{2.5}$ concentrations in the study domain, with the maximum increases exceeding
50% around Shijiazhuang and approximately 40% in Beijing and Tianjin, apparently higher
than the entire period averages. If for daytime mean, the percentage changes of PM$_{2.5}$ in the
above areas increased to 70% and 60%, respectively (figure not shown). It is striking that the
simulated maximum increase in hourly PM$_{2.5}$ concentration can be up to 372 μg/m$^3$ (186%)
in the vicinity of Shijiazhuang at about 10:00 LST on 24 February during the first haze
episode, which demonstrates the substantial impact of the radiative feedback on PM$_{2.5}$
concentration and haze formation.
It is worthwhile to further explore the effect of aerosol feedback during haze evolution.
We divided haze episode into three stages, the growth stage is defined as the time period of
PM$_{2.5}$ increase from clean condition to heavy pollution level, the persistence stage means the
duration period of haze and the dissipation stage means the period with a sharp decrease in
PM$_{2.5}$ concentration usually along with a cold front passage. During the first heavy haze
episode (20-26 February) in Beijing, aerosol radiative feedback caused the increases in PM$_{2.5}$
concentration of 55 μg m$^{-3}$, 84 μg m$^{-3}$ and 40 μg m$^{-3}$, with the fractional changes of 31%, 41%
and 67%, respectively, during the growth, persistence and dissipation stages. The larger
fractional change of PM$_{2.5}$ in the dissipation stage is due to the relatively large
feedback-induced increase and the lowest PM$_{2.5}$ concentration in the NoAer case in this stage.
During the second haze episode (1-4 March), the increases in PM$_{2.5}$ concentration due to
aerosol feedback were 25 μg m$^{-3}$, 45 μg m$^{-3}$ and 24 μg m$^{-3}$, with the fractional changes of
21%, 35% and 34%, respectively, which are lower than the feedback effect during the first





haze episode. So, in terms of magnitude, the largest feedback effect on PM$_{2.5}$ occurred in the
persistence stage, followed by that in the growth stage, although the fractional change of
PM$_{2.5}$ was larger in the dissipation stage.

Table 5 summarized the average feedback-induced changes in meteorological variables

and PM$_{2.5}$ concentrations over the BTH region during the entire and the first haze periods.
During the study period, due to the radiative feedback by all aerosols (FULL minus NoAer),
surface air temperature and wind speed decreased by 1.4 °C and 0.04 m s$^{-1}$, respectively, with
RH increased by 8.7% in the BTH. PBL height was reduced by 160 m (or a percentage
change of -18.6%) on average, along with a reduction of 3.3 m$^2$ s$^{-1}$ (-27.0%) in vertical
diffusivity coefficient (K$_z$), resulting in an increase of PM$_{2.5}$ level by 20.0 μg m$^{-3}$ (28.6%). It
is noticed that the above changes were strengthened during the severe haze episode on 20-26
February, with the 7-day average decreases in T2, WS10, PBL height and K$_z$ being up to -1.8
°C , -0.5 m s$^{-1}$, -183.6 m (-31.0%) and 3.9 m$^2$ s$^{-1}$ (-48.8%), respectively, and the PM$_{2.5}$
concentration increased by 45.1 μg m$^{-3}$ with a percentage increase of 38.7%. Because
aerosols affect solar radiation in daytime, in term of daytime mean, the 7-day mean changes
in T2, WS10 and PBL height were estimated to be -2.5 °C , -0.6 m s$^{-1}$ and -307.3 m (-37.6%),
respectively, leading to an increase of 49.3 μg m$^{-3}$ (48.5%) in PM$_{2.5}$ concentration.

The impact of aerosol radiative feedback in Beijing (Table 6) was stronger than the

regional mean. During the first haze episode, the 7-day average changes in T2, WS10, RH2,
PBL and PM$_{2.5}$ were estimated to be -2.1 °C, -0.6 m s$^{-1}$, 17.0%, -195.6 m (-35.9%) and 68.0
μg m$^{-3}$ (39.1%), respectively, and the daytime mean change in PM$_{2.5}$ concentration increased
to 83.2 μg m$^{-3}$ (60%), respectively.

Table 7 presents the average changes in major aerosol components (BC, sulfate and

nitrate) in PM$_{2.5}$ induced by the feedback effect. Over the BTH region, the feedback caused
the average increases in sulfate and nitrate by 5.0 μg m$^{-3}$ (46.4%) and 6.8 μg m$^{-3}$ (37.3%),
respectively, for the entire period, and by up to 12.6 μg m$^{-3}$ (66.9%) and 14.6 μg m$^{-3}$ (40.9%),
for the first haze episode. The feedback-induced increases in BC was 0.9 μg m$^{-3}$ (25.1%) and
1.9 μg m$^{-3}$ (32.9%), respectively, for the entire period and the first haze episode. It was
noticed that the feedback-induced changes in sulfate and nitrate concentrations were larger
than that in BC concentration. This was because that the concentrations of secondary aerosols





were increased not only by weakened vertical diffusivity but also by enhanced chemical
reactions due to the radiative feedback, which will be discussed in detail in section 5.2.
The above analysis demonstrates a significant impact of aerosol feedback on $PM_{2.5}$
concentration during winter haze episodes in the BTH region. Previous modeling studies
reported different degrees of aerosol radiative feedback in east China. Gao et al. (2015)
simulated an increase of near surface $PM_{2.5}$ concentrations to be 10-50 $\mu g\ m^{-3}$ or 5-25% in the
BTH during a severe haze episode on 10-15 January 2013 by using WRF-Chem. For the
similar time period and region, Wang et al. (2014a) reported an increase in $PM_{2.5}$
concentrations by 15-50 $\mu g\ m^{-3}$ or 10-30% by using a regional coupled model NAQPMS. Wu
et al. (2019) used WRF-Chem to investigate a haze episode from 5 December 2015 to 4
January 2016 in the North China Plain and found that the aerosol radiative effects can
enhance near-surface $PM_{2.5}$ concentration by 10.2 $\mu g\ m^{-3}$ (7.8%) on average.
The results from this study demonstrate a stronger aerosol-radiation feedback  than
previous modeling studies, with an average increase in $PM_{2.5}$ concentration by up to 45.1 $\mu g$
$m^{-3}$ (38.7%) during a severe haze episode and further to 49.3 $\mu g\ m^{-3}$ (48.5%) for daytime
mean over the BTH region. This study also highlights that the aerosol feedback effect can
result in an increase of hourly $PM_{2.5}$ concentrations by up to 372 $\mu g\ m^{-3}$ (186%) in the
vicinity of Shijiazhuang during the severe haze episode. The stronger feedback effect in this
study than previous model simulations is mainly due the predicted higher concentration of
aerosol components (especially inorganic aerosols) and aerosol optical properties, which are
also in a better agreement with observations. It is noticed that a recent study (Zhong et al.,
2018a) reported that the aerosol feedback effect contributed over 70% to $PM_{2.5}$ increase
during the cumulative explosive stage of haze event in winter Beijing based on integrated
analysis of observations from 2013-2016, which suggested a dominant role of the feedback
effect in haze formation.

**5 Process analysis of haze evolution and aerosol radiative feedback**
The process analysis (PA) method calculates the Integrated Process Rates (IPRs) and is
applied to quantify the individual contributions of different physical and chemical processes



to variations of PM$_{2.5}$ and its chemical components. These processes include emission,
horizontal and vertical advections (HADV and VADV), horizontal and vertical diffusions
(HDIF and VDIF), dry deposition (DDEP), cloud (CLD, including aqueous chemistry and
wet scavenging), gas chemistry (GAS), thermodynamic chemistry (Thermo) and
heterogeneous chemistry (HET). The focus of this study is Beijing, so the model grid cell
near the surface having Beijing is selected for analysis.

5.1 The mechanism of haze evolution related to various processes
5.1.1 Haze evolution during 20-26 February

There was a severe haze event lasting for about 7 days, with the maximum hourly PM$_{2.5}$

up to 482 µg m$^{-3}$ on 26 February. This haze was initially formed on 20 February, with the
observed surface PM$_{2.5}$ concentration less than 50 µg m$^{-3}$ on 19 February, rapidly increased to
343 µg m$^{-3}$ on 20 February, and reached 482 µg m$^{-3}$, followed by rapid haze dissipation on 26
February due to the arrival of a cold front.

PA was used to provide insights into the evolution mechanism of the haze episode, which

was divided into the clean, growth, persistence and dissipation stages in this study. Figure 8
shows the average process budgets for changes in PM$_{2.5}$ (which is the sum of sulfate, nitrate,
ammonium, BC, OC, SOC and primary PM$_{2.5}$) and its major components in Beijing during
the four stages of the first haze period (Figure 8) from the FULL simulation. Figure 8a shows
the hourly IPRs of PM$_{2.5}$ by physical and chemical processes. The emission of primary
aerosols was the largest contributor to the PM$_{2.5}$ mass with a constant IPR of 29.8 µg m$^{-3}$ h$^{-1}$
(not shown in Figure 8a for clarity) due to the use of a monthly based emission inventory.
Chemical processes (GAS, Thermo and HET) also contributed largely to PM$_{2.5}$, with
generally larger contributions in the growth and persistence stages. Thermodynamic
equilibrium processes and gas chemistry accounted for over 2/3 of the chemical contributions,
with the former process mainly accounting for the formation of nitrate and ammonium and
the latter one for sulfate formation. The contribution from heterogeneous reactions was
generally small, but when conditions were favorable (such as high RH and high aerosol
concentration providing sufficient reaction surfaces), its contribution would also be
significant, such as on the morning of 22 February, at nighttime from 23 to 24 February, and



on the mornings of 25 and 26 February. Vertical diffusion and dry deposition consistently
removed $PM_{2.5}$ from the atmosphere. In general, the larger IPRs from both VDIF and DDEP
during the clean and dissipation stages resulted in lower $PM_{2.5}$ concentrations, whereas the
lower IPRs from VDIF and DDEP in the growth stage favored aerosol accumulation. In the
persistence stage, the IPRs of VDIF and DDEP were generally small. It should be noted that
on every midday, when PBL was fully developed, the vertical diffusion reached the daily
maximum, producing distinctly large negative IPRs of VDIF. Advections (HADV and VADV)
and horizontal diffusion either contributed to the accumulation or loss of $PM_{2.5}$. During this
severe haze episode, horizontal diffusion served as a sink of $PM_{2.5}$, producing a negative IPR
of HDIF through the event. Horizontal advection served as a sink of $PM_{2.5}$ in most of the time,
leading to a negative IPR of HADV, however, when the removal of $PM_{2.5}$ by vertical
diffusion was strong at the midday, aerosols were advected to Beijing from surrounding areas
due to mass balance, resulting in a positive IPR of HADV. The positive IPR of VADV during
the growth and persistence stages of this event indicated that the downward transport of
aerosols from upper levels also contributed to the $PM_{2.5}$ increase, such as on the mornings of
22 and 25 February. In general, the IPRs (represented the net effect of all processes, denoted
by the red line in Figure 8a) exhibited small positive values from evening to next morning on
every day, indicating a gradually increasing $PM_{2.5}$ concentration, whereas on every midday,
relatively large negative IPRs occurred, indicating an apparent decrease in $PM_{2.5}$
concentration at that time. It should be mentioned that even in the persistence stage, the
diurnal variation of $PM_{2.5}$ occurred although the change rates were generally weaker than
those in the growth and dissipation stages.
Figure 8b to 8f show the mean IPRs for $PM_{2.5}$ and its major chemical components as well
as the key meteorological variables averaged over each stage to help interpret the formation
and evolution mechanism of this severe haze episode.
In the clean stage, emission and chemistry were the two major processes for $PM_{2.5}$
production (Figure 8b). Emission contributed predominately to $PM_{2.5}$ production (IPRs of
29.8 $\mu g\ m^{-3}\ h^{-1}$), whereas the contributions of gas (9.2 $\mu g\ m^{-3}\ h^{-1}$) and thermodynamic
chemistry (7.3 $\mu g\ m^{-3}\ h^{-1}$) were comparable. The most influential process for $PM_{2.5}$ removal
was vertical diffusion, with the IPRs of -30.3 $\mu g\ m^{-3}h^{-1}$, comparable to that of emission. Dry





deposition was the second most important process for PM$_{2.5}$ loss (-12.2 µg m$^{-3}$ h$^{-1}$), followed
by horizontal diffusion. Advection had a negligible effect on PM$_{2.5}$ in this stage. In the growth
stage, it is noteworthy that the contributions from vertical diffusion (VDIF) and dry
deposition (DDEP) to PM$_{2.5}$ removal decreased markedly from -30.3 µg m$^{-3}$ h$^{-1}$ and -12.2 µg
m$^{-3}$ h$^{-1}$ in the clean stage to -21.6 µg m$^{-3}$ h$^{-1}$ and -9.2 µg m$^{-3}$ h$^{-1}$, respectively (Figure 8b),
mainly due to the decrease in wind speed and the increase in stability indicated by the
reduced vertical diffusivity coefficient K$_z$ (Figure 8f), leading to increases in concentrations
of all species. It is impressive that the contributions from chemical processes
(GAS+Thermo+HET) increased apparently compared with those in the clean stage, with the
IPRs from gas, thermodynamic and heterogeneous chemistry increase to 12.1 µg m$^{-3}$ h$^{-1}$, 16.0
µg m$^{-3}$ h$^{-1}$ and 5.4 µg m$^{-3}$ h$^{-1}$, respectively. The increase in the contribution from
heterogeneous chemistry was mainly attributed to the increase in relative humidity and
aerosol surfaces, upon which heterogeneous reactions took place. It is noticed that the
contribution of thermodynamic chemistry increased with increasing relative humidity as well
along with haze formation (Figure 8f). The increase in the contribution of thermodynamic
chemistry was remarkable (with IPR from 7.3 to 16 µg m$^{-3}$ h$^{-1}$), because gas precursors of
aerosols increased due to weakened vertical diffusivity and higher relative humidity during
haze period favored condensation from gas to aerosol phase. It is of interest that vertical
advection also contributed to PM$_{2.5}$ production (IPR of 5.4 µg m$^{-3}$ h$^{-1}$) in this stage, which
indicated a potential downward import of PM$_{2.5}$ from upper layer. It is also noticed that
horizontal advection contributed to PM$_{2.5}$ loss (-12.8 µg m$^{-3}$ h$^{-1}$). This is because the strong
gradient between the increased PM$_{2.5}$ level in Beijing caused by weakened vertical diffusivity
and the relatively lower PM$_{2.5}$ level in the surrounding areas, which led to an outflow of
PM$_{2.5}$. In the growth stage, the net variation rate (IPR) of PM$_{2.5}$ concentration was 14.1 µg
m$^{-3}$ h$^{-1}$, in which emissions, chemical processes (GAS+Therm+HET) and physical processes
(HADV+VADV+HDIF+VDIF+DDEP) contributed 29.8 µg m$^{-3}$ h$^{-1}$, 33.5 µg m$^{-3}$ h$^{-1}$ and -49.2
µg m$^{-3}$ h$^{-1}$, respectively. In the persistence stage, chemical production rate of PM$_{2.5}$ changed
slightly, and the production and loss rates of PM$_{2.5}$ were similar, leading to an approximately
zero IPR in this stage (Figure 8b). In the dissipation stage, the contribution of vertical
diffusion and dry deposition to PM$_{2.5}$ loss increased largely, while the total chemical



production rate decreased, which resulted in a net IPR of -34.8 μg m$^{-3}$ h$^{-1}$, indicating a
substantial decrease in PM$_{2.5}$ concentration (Figure 8b). It was also noticed that HADV
contributed to PM$_{2.5}$ production in this stage, which was due to mass import to Beijing from
upwind areas by northwesterlies.
It should be mentioned that the contribution of emission was unchanged because the
monthly based emission inventory from MEIC was used, and the contribution of cloud
process was generally negligible throughout the period because there was little cloud and
precipitation during the study period.
We further use PA to interpret evolution processes of primary (BC) and secondary
(sulfate and nitrate) aerosols.
Black carbon is considered to be inert and chemical inactive, so it is governed solely by
physical processes. In the clean stage, BC production was contributed solely by emission (5.7
μg m$^{-3}$ h$^{-1}$), whereas vertical diffusion and dry deposition contributed equally to BC loss (-2.7
μg m$^{-3}$ h$^{-1}$), and other processes were negligible (Figure 8c). In the growth stage, the
contribution of vertical diffusion and dry deposition to BC loss decreased to -2.0 μg m$^{-3}$ h$^{-1}$
and -1.7 μg m$^{-3}$ h$^{-1}$, respectively, and the net rate of change was 0.7 μg m$^{-3}$ h$^{-1}$, indicating a
rapid increase of BC concentration in this stage (Figure 8c). In the persistence stage, the loss
rate by vertical diffusivity and dry deposition further increased mainly due to the increased
BC concentration (Figure 8c). It is noticed that horizontal advection somewhat contributed to
the loss of BC (-0.7 μg m$^{-3}$ h$^{-1}$), which indicated an increasing outflow of BC to surrounding
areas. The IPR was near zero, indicating a balance of production and loss rate in this stage. In
the dissipation stage, BC loss via vertical diffusion and dry deposition processes increased
largely, mainly due to increasing wind speed and vertical diffusivity, and the net IPR became
-1.6 μg m$^{-3}$ h$^{-1}$. This absolute value was larger than that in the growth stage (0.7 μg m$^{-3}$ h$^{-1}$),
which indicated a faster decrease in BC concentration than the BC increase in the growth
stage (Figure 8c).
As for secondary aerosols, like sulfate, contribution from direct emission was near zero.
In the clean stage, gas chemistry (5.9 μg m$^{-3}$ h$^{-1}$) was the predominant process for sulfate
production, and vertical diffusion contributed most to the loss (-5.2 μg m$^{-3}$ h$^{-1}$) (Figure 8d). In
the growth stage, contribution from vertical diffusion was reduced to -3.9 μg m$^{-3}$ h$^{-1}$ mainly



due to the decreased vertical diffusivity (Figure 8f), whereas positive contribution from gas
chemistry increased to 6.6 μg m$^{-3}$ h$^{-1}$, which was resulted from competitive processes. For
sulfate formation from gas chemistry ($SO_2+OH\rightarrow H_2SO_4$), the oxidation of $SO_2$ to sulfate was
weakened because of decreasing OH radical due to increasing aerosol attenuation of solar
radiation, however, $SO_2$ increased due to weakened vertical diffusivity, leading to a slight net
increase of sulfate concentration compared with the clean stage. It is noteworthy that the
sulfate production rate from heterogeneous reactions increased to 2.7 μg m$^{-3}$ h$^{-1}$, mainly due
to the increases in $SO_2$, aerosol surfaces and RH (as well as aerosol water content). All the
processes led to a net sulfate production rate of 2.7 μg m$^{-3}$ h$^{-1}$, in which chemistry played a
predominant role (IPR of 9.3 μg m$^{-3}$ h$^{-1}$). In the persistence stage, the contribution of gas and
heterogeneous processes further increased to 7.4 μg m$^{-3}$ h$^{-1}$ and 4.3 μg m$^{-3}$ h$^{-1}$, indicating an
increasing sulfate production through chemical processes (Figure 8d). It is interesting to note
that vertical diffusion contributed more to sulfate loss than in the growth stage, which was
mainly due to the higher sulfate level than in the growth stage while vertical diffusivity
coefficients were almost the same. The net IPR in this stage was just 0.2 μg m$^{-3}$ h$^{-1}$, which
indicated an approximate balance of production and loss. In the dissipation stage, increasing
vertical diffusivity was the dominant process for sulfate loss, and chemical contribution
decreased. It is noticed a positive contribution to sulfate from horizontal advection (IPR of
4.3 μg m$^{-3}$ h$^{-1}$), which was due to an import of sulfate from upwind areas of Beijing by
northwesterly winds, like those for $PM_{2.5}$ and BC.
For nitrate, in the clean stage, thermodynamic process (4.5 μg m$^{-3}$ h$^{-1}$) was the largest
contributor to nitrate production (Figure 8e). During the growth stage, the contribution of
thermodynamic processes (10.2 μg m$^{-3}$ h$^{-1}$) increased by over a factor of two and was larger
than the contribution from heterogeneous process (Figure 8e). The substantial increase in the
contribution of thermodynamic processes to nitrate production was due to the combined
effects of the increased level of nitrate precursors ($HNO_3$ and $NH_3$) resulting from weakened
diffusivity and the increased RH favorable for gas to aerosol conversion. The contribution of
heterogeneous reactions increased as well due to the increased aerosol surface and relative
humidity. The net rate of nitrate change in this stage was 5.3 μg m$^{-3}$ h$^{-1}$. In the persistence
stage, the contribution from heterogeneous reactions changed slightly while the contribution





from thermodynamic process somewhat reduced (Figure 8e). This is because more $NH_3$ was
consumed to neutralize the increased sulfate, leaving less $NH_3$ to react with $HNO_3$, and thus
producing fewer nitrate. The near zero net IPR of nitrate in this stage also indicated a balance
of production and loss. In the dissipating stage, the contribution of chemical processes was
almost the same as that in the clean stage, while physical processes dominated the loss and
the net IPR of nitrate (Figure 8e).

5.1.2 Haze evolution during 1-4 March
We also investigate another haze period of 1-4 March using PA (Figure 9). The hourly
IPRs by different processes are shown in Figure 9a. An apparent difference between this
episode and the first one was the positive IPRs of HADV during this episode, especially in
the growth stage from 21:00 (LST) on 1 March to 9:00 (LST) on 2 March, which indicated
that horizontal transport contributed to the haze formation. Another difference is that the
chemical processes, especially heterogeneous reactions contributed less to the $PM_{2.5}$ mass
during the persistence stage, such as from 10:00 (LST) on 2 March to 3:00 (LST) on 4 March,
which will be discussed below.
The IPRs for $PM_{2.5}$ and its components and meteorological variables averaged over each
stage during this episode are calculated and presented in Figure 9b to 9f. For BC (Figure 9c),
the most evident difference from the first haze episode occurred in the growth stage, in which
horizontal advection contributed 1.5 $\mu g\ m^{-3}\ h^{-1}$ to BC production, which was comparable in
magnitude to the negative contributions from vertical diffusion and dry deposition (-1.3 $\mu g$
$m^{-3}\ h^{-1}$), suggesting the import of BC into Beijing from surrounding areas. The wind direction
in the south of Beijing at this stage was southerly and wind speed was about 2-3 $m\ s^{-1}$, so the
transport of pollutants from southern Hebei apparently contributed to the increase of BC level
in Beijing. Differently, during the first haze event on 20-26 February, wind direction was
easterly, bringing less polluted air mass from the Bohai Sea and northern Tianjin, so
horizontal advection contributed less to BC in Beijing. This transport feature was also
reflected in the change rates of sulfate (Figure 9d), nitrate (Figure 9e) and $PM_{2.5}$ (Figure 9b)
concentrations. An observational study for the same haze period in Beijing (Ma et al., 2017)
also suggested the important role of regional transport from the south of Beijing in haze



formation.

For sulfate (Figure 9d), although chemical processes still contributed most to sulfate

production in the growth stage (6.0 μg m$^{-3}$ h$^{-1}$), it is noticed that gas chemistry (5.9 μg m$^{-3}$ h$^{-1}$)
accounted for most of the sulfate production, whereas contribution from heterogeneous
reactions was smaller than that in the first haze episode mainly due to lower relative humidity.
In the growth stage, the net IPR was 1.9 μg m$^{-3}$ h$^{-1}$, 30% smaller than that for the first haze,
indicating a weaker secondary aerosol formation during this haze episode. In the persistence
stage, sulfate production from gas phase oxidation was almost balanced by the loss from dry
deposition and vertical diffusion, resulting in a net IRP of -0.1 μg m$^{-3}$ h$^{-1}$, indicating a small
variation of sulfate concentration during this stage on average.

For nitrate, in the growth stage, it is of interest to note that heterogeneous reactions (5.5

μg m$^{-3}$ h$^{-1}$) dominated over thermodynamic processes (2.7 μg m$^{-3}$ h$^{-1}$) in nitrate formation,
which could be due to the low RH in this stage. Fountoukis and Nenes (2007) indicated that
nitrate aerosol is hardly formed in the ISORROPIA II model when RH is below 40%. The
average RH is about 37% during this haze episode, resulting in more nitrate formed by
heterogeneous reactions. The net IPR in the growth stage was 3.7 μg m$^{-3}$ h$^{-1}$, approximately
30% smaller than that in the first haze episode. In the persistence stage when relative
humidity increased to 51%, nitrate formation via thermodynamic processes became important,
and due to competition, nitrate formation from heterogeneous reactions was reduced.

For PM$_{2.5}$ (Figure 9b), in the growth stage, the IPR of PM$_{2.5}$ concentration was 13.0 μg

m$^{-3}$ h$^{-1}$, in which emission, chemical processes (GAS+Therm+HET) and physical processes
(HADV+VADV+HDIF+VDIF+DDEP) contributed 29.8 μg m$^{-3}$ h$^{-1}$, 23.9 μg m$^{-3}$ h$^{-1}$ and -40.7
μg m$^{-3}$ h$^{-1}$, respectively. It is noteworthy that horizontal advection process (HADV)
contributed 22.4 μg m$^{-3}$ h$^{-1}$ to PM$_{2.5}$ production in this episode, which was comparable to the
total chemical production of 23.9 μg m$^{-3}$ h$^{-1}$. This reveals the comparable contributions to
PM$_{2.5}$ in Beijing from local sources and regional transport during this haze episode. In the
persistence stage, because of the change in wind direction and lower wind speed, the regional
transport of PM$_{2.5}$ became weak. The IPRs were -4.0 μg m$^{-3}$ h$^{-1}$ for HADV and 1.2 μg m$^{-3}$ h$^{-1}$
for VADV, respectively, which were obviously smaller than those in the first haze episode. In
the dissipation stage, physical processes except HADV all contributed to the loss of PM$_{2.5}$.



Compared with the first haze episode, the negative IPR of VADV decreased mainly due to the larger wind speeds in this episode, as more $PM_{2.5}$ was removed by VADV, the remaining $PM_{2.5}$ loss by vertical diffusion decreased, consequently a weakened VDIF. The positive IPR of HADV increased as well due to larger wind speed than that in the first episode in this stage.

The above process analyses reveal that for the first haze episode (20-26 February) in Beijing, local emissions and chemical processes were the main contributors to the formation and persistence of the haze pollution. However, for the second haze (1-4 March), regional transport or horizontal advection played a more important role in haze formation, with a similar magnitude to local emissions and chemical productions in the growth stage. In all, for both episodes, local emission, chemical reaction and horizontal advection were major processes contributing to $PM_{2.5}$ increase, whereas vertical processes (diffusion, dry deposition and advection) were major processes for $PM_{2.5}$ removal. As the pollution level increased, the contribution of secondary aerosols through chemical formation to $PM_{2.5}$ increased apparently in Beijing.

5.2 Contributions of physical and chemical processes to the aerosol feedback

5.2.1 The first haze episode (20-26 February)

Figure 10 shows the contributions of each process to the feedback-induced difference in the change rates of $PM_{2.5}$ and its major components (ΔIPR) during the first haze episode (20-26 February), which were derived from the difference between cases with and without aerosol radiative effects (FULL minus NoAer).

The definition of the four stages during haze evolution is the same as that in section 5.1.1. For BC (Figure 10b) in the clean stage, the aerosol feedback caused a decrease in vertical diffusion and advection (Figure 10e), leading to an increase in BC concentration with the ΔIPR of 0.40 μg m$^{-3}$ h$^{-1}$ from VDIF+VADV, concurrently, the feedback caused an increased loss of BC through horizontal diffusion (HDIF) and advection (HADV) and dry deposition (DDEP) due to the increased BC concentration, with the ΔIPR of -0.39 μg m$^{-3}$ h$^{-1}$ from HADV+HDIF+DDEP (Figure 10b). The net ΔIPR was near zero, which indicated a negligible feedback effect during the clean stage. In the growth stage, the feedback caused a



pronounced decrease in vertical diffusivity, advection, as well as dry deposition velocity,
leading to apparent increases in BC level, with the contributions to ΔIPRs from VDIF, VADV,
and DDEP being 0.50 μg m$^{-3}$ h$^{-1}$, 0.50 μg m$^{-3}$ h$^{-1}$ and 0.20 μg m$^{-3}$ h$^{-1}$, respectively (Figure
10b). The increase in BC concentration consequently led to an increase in outflow via HADV
and HDIF, with the ΔIPRs of -0.63 μg m$^{-3}$ h$^{-1}$ and -0.12 μg m$^{-3}$ h$^{-1}$, respectively, which tended
to reduce BC concentration. The total effect by summing the processes exhibited a net
positive ΔIPR of 0.44 μg m$^{-3}$ h$^{-1}$, which indicated an apparent increase in BC concentration
due to the feedback. In the persistence stage, the sign of ΔIPR for each process was the same
as that in the growth stage, and the ΔIPR by vertical processes (0.84 μg m$^{-3}$ h$^{-1}$ from
VADV+VDIF+DDEP) was generally balanced by that of horizontal processes (-0.80 μg m$^{-3}$
h$^{-1}$ from HADV+HDIF) and led to a net ΔIPR of 0.04 μg m$^{-3}$ h$^{-1}$ (Figure 10b), which
indicated the difference in the BC change rate between the FULL and NoAer cases was small
in this stage. In the dissipating stage, the ΔIPRs were negative for all the processes except for
HADV. This was because of the higher BC levels due to the feedback, which caused more
BC to be removed than without feedback, although the vertical diffusion coefficient was
smaller due to the feedback. The positive ΔIPR from HADV suggested the enhanced BC
import into Beijing from upwind regions due to the feedback. The sum of these processes
produced a net ΔIPR of -1.20 μg m$^{-3}$ h$^{-1}$, which indicated a larger decreasing rate of BC
concentration (from haze to clean level) due to aerosol feedback in this stage.

For sulfate (Figure 10c), in the clean stage, the feedback-induced changes were as small

as those for BC. In the growth stage, besides the positive ΔIPRs by VDIF, VADV and DDEP
as those for BC, the most impressive feature was the larger contributions from GAS and HET,
with the ΔIPRs being 0.29 μg m$^{-3}$ h$^{-1}$ and 1.73 μg m$^{-3}$ h$^{-1}$, respectively, much larger than those
(0.11 μg m$^{-3}$ h$^{-1}$ and 0.23 μg m$^{-3}$ h$^{-1}$) in the clean stage because of the increased gas
precursors, aerosol surfaces and RH due to the feedback effect, which enhanced chemical
formation (Figure 10c, 10e). The sum of the ΔIPRs by all the processes was 1.92 μg m$^{-3}$ h$^{-1}$,
indicating an apparent increase in sulfate concentration due to the feedback effect. In the
persistence stage, the ΔIPRs by GAS and HET increased. However, the ΔIPR of VDIF
became negative, which could be explained by the increased sulfate concentration due to
aerosol feedback caused more sulfate to be removed through vertical diffusion, leading to a





negative ΔIPR of VDIF, although the vertical diffusion coefficient was reduced by the
feedback. In the dissipation stage, the ΔIPR by HET decreased because the feedback-induced
differences in the concentrations of precursors and aerosols became smaller. The large
negative ΔIPR by VDIF indicated a larger decreasing rate in sulfate concentration from the
persistence to clean stages due to the feedback.

For nitrate (Figure 10d), the feedback-induced IPR changes in the clean stage were

similar to those for sulfate. In the growth stage, remarkable increases in nitrate formation
from Thermo and HET processes occurred, with the ΔIPRs of 3.30 μg m$^{-3}$ h$^{-1}$ and 0.50 μg m$^{-3}$
h$^{-1}$, respectively (Figure 10d). The increased gas precursors and RH due to the aerosol
feedback reinforced chemical formation processes. In this stage, the overall ΔIPR was 3.90
μg m$^{-3}$ h$^{-1}$, suggesting a faster increasing rate in nitrate concentration in consideration of
aerosol feedback. In the persistence stage, the ΔIPR by Thermo was smaller than that in the
growth stage (Figure 10d). This could be explained that the apparent increase in sulfate
concentration via HET and GAS due to the feedback (Figure 10c) in this stage consumed
more ammonia, which inhibited the formation of nitrate ammonium via thermodynamic
processes. The net ΔIPR by all the processes in this stage was near zero, which indicated that
the radiative feedback exerted little effect on the change rate of nitrate concentration during
this stage. In the dissipation stage, the attenuation of solar radiation by aerosols was
weakened because of the decrease in aerosol concentration, meanwhile, the concentrations of
gas precursors (NO$_x$) were elevated due to the feedback, the combined effect resulted in an
increase of photochemical production of HNO$_3$; in addition, RH was increased due to the
feedback as well, as a result, nitrate formation via thermodynamic process was enhanced,
leading to a positive ΔIPR of 3.73 μg m$^{-3}$ h$^{-1}$ by Thermo in this stage.

For PM$_{2.5}$, the net ΔIPR due to aerosol feedback in the clean stage was 0.30 μg m$^{-3}$ h$^{-1}$, in

which 1.22 μg m$^{-3}$ h$^{-1}$ was from chemical processes (GAS+Thermo+HET) and -0.90 μg m$^{-3}$
h$^{-1}$ from physical processes (HADV+VADV+HDIF+VDIF+DDEP) (Figure 10a). In the
growth stage, the net ΔIPR was 9.50 μg m$^{-3}$ h$^{-1}$, which meant in every hour, approximate 9.50
μg m$^{-3}$ of PM$_{2.5}$ mass was elevated in Beijing due to the feedback effect. The above
feedback-induced difference in the change rate of PM$_{2.5}$ (ΔIPR) resulted from a combined
effect from chemical processes (7.27 μg m$^{-3}$ h$^{-1}$) and physical processes (2.23 μg m$^{-3}$ h$^{-1}$),



which suggested that chemical processes contributed more to the $PM_{2.5}$ increase than physical
processes. However, it was noted that the increased contribution from chemical processes
was related to increasing gas precursors, which was partly associated with physical processes.
It was noteworthy that the positive ΔIPRs were contributed by both chemical processes (GAS,
Thermo and HET) and vertical movements (VADV, VDIF and DDEP) (Figure 10a). The sum
of positive ΔIPRs was 22.88 μg $m^{-3}$ $h^{-1}$, in which 7.27 μg $m^{-3}$ $h^{-1}$ was from chemical
processes and 15.61 μg $m^{-3}$ $h^{-1}$ from vertical movements. This suggested a larger
feedback-induced $PM_{2.5}$ increase through vertical movements than via chemical processes.
However, the outflow (HADV+HDIF) of $PM_{2.5}$ was also enhanced due to the increased $PM_{2.5}$
level by aerosol feedback, producing a negative ΔIPR (-13.38 μg $m^{-3}$ $h^{-1}$), and partly
offsetting the positive ΔIPR (15.61 μg $m^{-3}$ $h^{-1}$) by vertical movements, resulting in a net ΔIPR
of 2.23 μg $m^{-3}$ $h^{-1}$ from all the physical processes. In the persistence stage, the sign of ΔIPRs
by different processes generally resembled those in the growth stage except that of VDIF
whose ΔIPR was negative, which indicated more removal though VDIF mainly due to the
increased secondary aerosol concentrations by aerosol feedback. The net ΔIPR by all the
processes was 0.40 μg $m^{-3}$ $h^{-1}$ in this stage, indicating a small influence of aerosol feedback
on the change rate of $PM_{2.5}$ concentration. In the dissipating stage (Figure 10a), the large
negative ΔIPR from VDIF indicated more $PM_{2.5}$ mass was removed via vertical diffusion
while considering aerosol feedback, although the feedback induced a smaller vertical
diffusivity coefficient. The net ΔIPR of -24.60 μg $m^{-3}$ $h^{-1}$ indicated a larger decreasing rate of
$PM_{2.5}$ concentration in the FULL case than in the NoAer case.

5.2.2 The second haze episode (1-4 March)
For BC in the second haze episode (1-4 March), the most obvious difference from the
first episode was in the growth stage, in which the ΔIPR by horizontal advection (HADV)
was 0.70 μg $m^{-3}$ $h^{-1}$ (Figure 11b). The radiative feedback led to a weakened vertical
diffusivity and a decreased PBL height (Figure 11e), which favored the accumulation of BC
and caused a positive ΔIPR of 0.40 μg $m^{-3}$ $h^{-1}$ from VDIF. The wind direction in the growth
stage was southerlies as discussed above, bringing aerosols from the south to Beijing. The
aerosol feedback enhanced BC concentration in source regions through reducing vertical





diffusivity, leading to an increased northward flux of BC and a positive ΔIPR from HADV.
The higher BC concentration due to the feedback via HADV and VDIF consequently led to
an increase in BC outflow out of Beijing via vertical advection (VADV) and horizontal
diffusion (HDIF), with the ΔIPRs of -0.60 μg m$^{-3}$ h$^{-1}$ and -0.20 μg m$^{-3}$ h$^{-1}$, respectively. In this
stage, the net ΔIPR of BC was 0.20 μg m$^{-3}$ h$^{-1}$, in which 0.50 μg m$^{-3}$ h$^{-1}$ was from horizontal
movements   (HADV+HDIF)   and   -0.30   μg   m$^{-3}$   h$^{-1}$   from   vertical   movements
(VADV+VDIF+DDEP), indicating that the feedback effect strengthened the contribution of
horizontal movements to surface BC concentration in Beijing. In the persistence stage (Figure
11b), the net ΔIPR was also near zero (-0.02 μg m$^{-3}$ h$^{-1}$), indicating that the BC change rate
was merely affected by the feedback in this stage. In the dissipation stage (Figure 11b), the
ΔIPRs were negative for all the processes except for VDIF. This could be attributed to the
higher BC levels due to the feedback, which caused more BC to be removed than without
feedback through these processes. The net ΔIPR was -0.17 μg m$^{-3}$ h$^{-1}$, the same as that in the
growth stage, but with opposite sign.

For sulfate (Figure 11c), in the growth stage, different from the relatively large positive

ΔIPR by chemical processes in the first haze episode, the feedback caused small IPR changes
via chemical production because SO$_2$ concentration in this episode was lower than that in the
first one and sulfate was mainly formed in upwind regions and transported to Beijing.
Consequently, relatively large sulfate increases through HADV and VDIF in this episode. In
this stage, the feedback caused a slight increase in sulfate concentration by GAS with ΔIPR
of 0.17 μg m$^{-3}$ h$^{-1}$ due to slightly elevated precursors, however, because of the low relative
humidity (mean RH was 38%) and competitive processes, heterogeneous reactions were
depressed. In terms of physical processes, due to the feedback effect, horizontal transport
(HADV) was strengthened (ΔIPR of 1.0 μg m$^{-3}$ h$^{-1}$) due to the increased sulfate concentration
to the south of Beijing, meanwhile, the weakened vertical diffusivity caused an increase in
sulfate concentration by VDIF and DDEP, with the ΔIPRs of 1.0 μg m$^{-3}$ h$^{-1}$ and 0.57 μg m$^{-3}$
h$^{-1}$, respectively, consequently, the outflow of sulfate out of Beijing was also increased via
vertical advection (VADV) and horizontal diffusion (HDIF). The net ΔIPR in the growth
stage was 0.90 μg m$^{-3}$ h$^{-1}$, indicating an apparent increase in sulfate concentration due to the
feedback. In the persistence stage, the ΔIPRs by GAS and HET changed slightly compared



with those in the growth stage. The negative ΔIPR by VDIF indicated more loss of sulfate by
vertical diffusion while considering aerosol feedback. The net ΔIPR in this stage was 0.02 μg
m$^{-3}$ h$^{-1}$, indicating a negligible feedback effect on sulfate change rate in this stage. In the
dissipation stage, the feedback-induced higher sulfate concentration caused more removal of
sulfate via physical processes except HADV, resulting in a net ΔIPR of -0.64 μg m$^{-3}$ h$^{-1}$. The
positive ΔIPR from HADV was due to the strengthened import from upwind areas due to the
feedback.
For nitrate, in the growth stage, the feedback also induced an increase in nitrate
concentration via horizontal advection like sulfate (Figure 11d). The increases in gas
precursors and aerosol surfaces due to the feedback enhanced nitrate formation, resulting in
nitrate increases via Thermo and HET, with the ΔIPRs of 0.88 μg m$^{-3}$ h$^{-1}$ and 0.46 μg m$^{-3}$ h$^{-1}$,
respectively. To the persistence stage, the chemical production of nitrate increased largely
caused by the feedback, with the ΔIPR of Thermo being 4.30 μg m$^{-3}$ h$^{-1}$. The reason could be
the low RH in the growth stage (38% shown in Figure 9f) left most of nitric acid remained in
gas phase together with the increase in RH due to the feedback (13.2% shown in Figure 11e)
drove its conversion from gas to aerosol phase. Due to the enhanced thermodynamics
production, nitrate formation via heterogeneous reactions was depressed in this stage. The
increased nitrate concentration via Thermo led to larger removal via vertical diffusion,
resulting in a negative ΔIPR of -4.80 μg m$^{-3}$ h$^{-1}$ by VDIF, and a net ΔIPR of -0.10 μg m$^{-3}$ h$^{-1}$.
In the dissipation stage, like that in the first haze episode, the reduced aerosol attenuation of
solar radiation and increased RH induced by aerosol feedback led to an increase in nitrate via
thermodynamic process, with the ΔIPR of 1.80 μg m$^{-3}$ h$^{-1}$ by Thermo. Consequently,
heterogeneous reactions were depressed due to competitive processes (ΔIPR of -0.97 μg m$^{-3}$
h$^{-1}$ by HET). In this stage, because of the higher nitrate concentration, the feedback led to
larger removal by vertical processes (the ΔIPR of VADV+VDIF+DDEP was -3.23 μg m$^{-3}$ h$^{-1}$),
with a net ΔIPR of -1.78 μg m$^{-3}$ h$^{-1}$, similar to the ΔIPR in the growth stage but with opposite
sign.
For PM$_{2.5}$ (Figure 11a), the net ΔIPR due to aerosol feedback in the growth stage was
2.40 μg m$^{-3}$ h$^{-1}$, with 1.40 μg m$^{-3}$ h$^{-1}$ from physical processes
(HADV+VADV+HDIF+VDIF+DDEP) and 1.0 μg m$^{-3}$ h$^{-1}$ from chemical processes





(GAS+Thermo+HET), which indicated that the feedback-induced increase in $PM_{2.5}$
concentration per hour was produced through larger contributions from physical processes
than chemical processes in this episode. HADV contributed most to the $PM_{2.5}$ increase (with
ΔIPR of 10.20 μg m$^{-3}$ h$^{-1}$), followed by VDIF (with ΔIPR of 2.90 μg m$^{-3}$ h$^{-1}$). As mentioned
above, the weakened vertical diffusivity caused by the feedback enhanced aerosol
concentrations in the entire BTH region, meanwhile, the feedback induced a southeast wind
anomaly with a slight change in wind speed in the regions south of Beijing. The combined
effect of the elevated aerosol concentrations and southeast wind anomaly brought more
aerosols to Beijing. In the persistence stage, the feedback increased $PM_{2.5}$ concentration
mainly through chemical processes, with the ΔIPR of 6.05 μg m$^{-3}$ h$^{-1}$, which was mainly
resulted from the enhanced thermodynamic production of ammonium nitrate, and such
increase in aerosol mass due to feedback led to more aerosols to be diffused than that without
feedback, leading to the ΔIPR of -7.30 μg m$^{-3}$ h$^{-1}$ by VDIF. It is noticed that the signs of the
ΔIPRs by VDIF were opposite between the growth and persistence stages even though the
vertical diffusivities were both decreased. In the growth stage, the $PM_{2.5}$ concentration was
gradually increasing, the effect of the weakened vertical diffusivity was dominated, resulting
in a positive ΔIPR by VDIF which favored further accumulation of aerosols; in the
persistence stage, the aerosol concentration had already been elevated to a high level, the
effect of higher concentration surpassed that of weakened vertical diffusivity due to the
feedback and led to a negative ΔIPR, which meant the feedback caused more loss of $PM_{2.5}$
via VDIF. In the persistence stage, the net ΔIPR was 0.44 μg m$^{-3}$ h$^{-1}$, in which -5.6 μg m$^{-3}$ h$^{-1}$
from physical processes and 6.05 μg m$^{-3}$ h$^{-1}$ from chemical processes, which indicated the
feedback-induced overall changes in the change rate of $PM_{2.5}$ concentration in this stage were
relatively small. In the dissipating stage, the removal of $PM_{2.5}$ was enhanced by the feedback
through all the processes except HADV mainly due to the increased $PM_{2.5}$ concentration, the
positive ΔIPR by HADV was caused by the enhanced import from upwind areas due to the
feedback. In this stage, the feedback effect enhanced the removal of $PM_{2.5}$, which was
reflected by the net negative ΔIPR of -4.30 μg m$^{-3}$ h$^{-1}$.
The above analyses quantify the key processes contributing to the aerosol radiative
feedback in Beijing during the two haze episodes. In the growth stage of the first haze



episode, the feedback-induced $PM_{2.5}$ enhancement was attributed to the positive contributions
from chemical processes and vertical movements, but partly offset by the increased outflow
of $PM_{2.5}$ via horizontal advection, resulting in a larger increase in $PM_{2.5}$ through chemical
processes than that from physical processes. Differently, duirng the second haze episode, the
feedback-induced $PM_{2.5}$ enhancement in the growth stage was larger by physical processes
than that by chemical processes, and horizontal advection contributed most to the $PM_{2.5}$
enhancement. In all, the radiative feedback increased the cumulative rate of aerosols in the
growth stage via promoting chemical formations, weakening vertical diffusions and/or
enhancing regional transport by horizontal advection. For both episodes, the radiative
feedback exerted small effect on the change rate of $PM_{2.5}$ concentration during the persistence
stage and reinforced the decreasing rate of $PM_{2.5}$ in the dissipation stage.

## 1148    6 Conclusion

Several severe haze events occurred in the winter of 2014, with the most severe one on
20-26 February. An online-coupled regional atmospheric chemistry/aerosol-climate model
(RIEMS-Chem) was developed and utilized to investigate the mechanisms of haze formation
and aerosol radiative feedback in the Beijing-Tianjin-Hebei (BTH) region. The heterogeneous
chemical reactions were treated in the model and the measured size distribution and mixing
state of aerosols in Beijing were used to constrain the model. Two numerical experiments,
with and without aerosol effects were conducted to explore the aerosol radiative effects
(AREs) and feedbacks on meteorological fields and aerosol distributions. Processes analysis
technique was implemented in RIEMS-Chem to quantify the individual contributions from
various physical and chemical processes to aerosol evolution and radiative feedback. Model
performance was comprehensively evaluated by comparing with a variety of observations for
meteorological variables, surface shortwave radiation, PBL heights, $PM_{2.5}$ and its chemical
components, as well as aerosol optical properties in the BTH region. The comparisons
demonstrated that RIEMS-Chem was able to represent the magnitudes and variations of the
above variables reasonably well, in particular, improving the simulation of inorganic aerosols
and AOD, which was often underpredicted in current on-line coupled models. It is
encouraging that by considering the aerosol radiative effects, the model apparently improved
predictions for meteorological variables, $PM_{2.5}$ and its chemical compositions and aerosol
optical properties in the BTH region, suggesting the importance and necessity for developing
chemistry-climate online coupled models in both air quality and climate research.
During the study period, the meteorological conditions were characterized by weak
southerly winds, high RH and low PBL height, which favored aerosol accumulation and haze
formation in the BTH region. The average T2, WS10, RH2, PBL height and $PM_{2.5}$
concentration from the FULL case were simulated to be 0.6 °C, 1.2 m s$^{-1}$, 67.0%, 698.4 m
and 90.0 μg m$^{-3}$, respectively, over the BTH region during the study period.
The distribution pattern of AOD generally resembled that of $PM_{2.5}$, with the domain
mean value of 0.78 and the maximum up to 1.1 during the study period. It was noteworthy
that the simulated SSA averaged over the BTH region and the study period was 0.91, which
indicated the dominance of scattering aerosols. The domain and period average AREs at the
surface, in the atmosphere and at the TOA were estimated to be -37 W m$^{-2}$, 19 W m$^{-2}$ and -18
W m$^{-2}$, respectively, and they were enhanced to -57 W m$^{-2}$, 25 W m$^{-2}$ and -32 W m$^{-2}$ during
the most severe haze episode (20-26 February). It was striking that the maximum hourly
AREs at the surface and at TOA reached -384 W m$^{-2}$ and -231 W m$^{-2}$ around noon time in the
vicinity of Shijiazhuang during the first haze episode. The magnitude of the model simulated
AREs during the haze episode in this study agreed favorably with previous observational
based estimates.
The aerosol radiative effects generally led to a reduction in surface air temperature in the
entire domain with larger decrease in southern BTH (-1.2 °C ~ -2 °C), accompanied by an
increase in RH2 (10% ~ 16%) and a decrease in PBL height (-240 m ~ -210 m). The changes
in these meteorological variables were strengthened during the severe haze episode.
Noticeably, $PM_{2.5}$ concentrations were consistently increased over the BTH region due to the
aerosol feedback, with the maximum average increase exceeding 33 μg m$^{-3}$ (33%) in southern
Hebei and portions of Beijing and Tianjin during the study period, and the maximum hourly
increase was up to 372 μg m$^{-3}$ (186%) in the vicinity of Shijiazhuang during the severe haze
episode. In terms of domain and period average, the feedback-induced changes were -1.4 °C
for T2, -0.04 m s$^{-1}$ for WS10, 8.7% for RH2, -3.3 m$^2$ s$^{-1}$ for vertical diffusion coefficient,





-160.0 m (-19%) for PBL height and 20.0 μg m$^{-3}$ (29%) for PM$_{2.5}$ concentration. The
magnitude of the above changes were enhanced during the severe haze episode, with the
7-day mean changes in T2, WS10, RH2, PBL height and PM$_{2.5}$ concentration being -1.8 °C,
-0.5 m s$^{-1}$, 9.8%, -183.6 m (-31%) and 45.1 μg m$^{-3}$ (39%), respectively, which demonstrated
the significant aerosol radiative feedback on PM$_{2.5}$ accumulation and haze formation. The
changes in sulfate and nitrate concentrations were larger than that in BC concentration
because secondary aerosols were increased not only by weakened vertical diffusivity but also
by enhanced chemical reactions caused by the feedback.
The magnitude of the feedback effect varied remarkably during haze evolution. The
absolute change in PM$_{2.5}$ concentration caused by the feedback was largest in the persistence
stage, followed by those in the growth stage and in the dissipating stage. In Beijing, the
feedback-induced increases in PM$_{2.5}$ concentration were 55 μg m$^{-3}$, 84 μg m$^{-3}$, 40 μg m$^{-3}$,
respectively, during the growth, persistence and dissipation stages of the severe haze episode.
PA method was applied to calculate the IPRs for quantifying the individual contributions
from physical and chemical processes to variations of PM$_{2.5}$ and its chemical components
during haze episodes in Beijing. Two haze episodes were analyzed and compared to elucidate
the mechanism of haze formation and evolution. For the first haze episode, the net IPR for
PM$_{2.5}$ was 14.1 μg m$^{-3}$ h$^{-1}$ in the growth stage, in which emissions, chemical processes and
physical processes contributed 29.8 μg m$^{-3}$ h$^{-1}$, 33.5 μg m$^{-3}$ h$^{-1}$ and -49.2 μg m$^{-3}$ h$^{-1}$,
respectively, which indicated a remarkable PM$_{2.5}$ increase contributed by chemical processes
in this stage. The most influential processes for PM$_{2.5}$ loss and production were vertical
diffusion and thermodynamic processes, respectively. Compared with the clean stage, the
losses by vertical diffusion and dry deposition reduced largely, and the production by
chemical processes increased, both leading to an evident increase in surface PM$_{2.5}$
concentrations in the growth stage. In the persistence stage, the production and loss of PM$_{2.5}$
were almost equal, resulting in an approximately zero IPR in this stage. In the dissipation
stage, the loss of PM$_{2.5}$ by vertical diffusion and dry deposition increased greatly, leading to a
net IPR rate of -34.8 μg m$^{-3}$ h$^{-1}$, which meant a substantial decrease in PM$_{2.5}$ concentration.
For the second haze episode, the net IPR for PM$_{2.5}$ was 13.0 μg m$^{-3}$ h$^{-1}$ in the growth
stage, in which emissions, chemical processes and physical processes contributed 29.8 μg m$^{-3}$



$h^{-1}$, 23.9 µg $m^{-3}$ $h^{-1}$ and -40.8 µg $m^{-3}$ $h^{-1}$, respectively. It was noteworthy that the contribution
of horizontal advection to $PM_{2.5}$ was of a similar magnitude to the contributions from local
emissions and chemical processes, with the mean IPR of 22.4 µg $m^{-3}$ $h^{-1}$, which indicated the
important contribution of regional transport to haze formation in Beijing. Process analysis for
the changes in $PM_{2.5}$ components during haze evolution was also conducted.

The contribution of each physical and chemical process to the feedback-induced changes

in $PM_{2.5}$ and its major components were explored and quantified. For the first haze episode,
the fast increase in $PM_{2.5}$ (ΔIPR of 9.5 µg $m^{-3}$ $h^{-1}$) due to aerosol feedback in the growth stage
was mainly attributed to the changes in vertical movements (VDIF and VADV) and chemical
processes, but the increased outflow via horizontal advection (HADV) partly offset the
increased $PM_{2.5}$ due to vertical movements, which caused a larger contribution to the $PM_{2.5}$
increase from chemical processes (ΔIPR of 7.27 µg $m^{-3}$ $h^{-1}$) than that from physical processes
(ΔIPR 2.23 µg $m^{-3}$ $h^{-1}$). However, during the second haze episode, the feedback-induced
$PM_{2.5}$ increase (ΔIPR of 2.4 µg $m^{-3}$ $h^{-1}$) in the growth stage was mainly contributed by
physical processes (ΔIPR of 1.40 µg $m^{-3}$ $h^{-1}$) rather than that by chemical processes (ΔIPR of
1.0 µg $m^{-3}$ $h^{-1}$), and among physical processes, the $PM_{2.5}$ increase was mainly attributed to the
increased horizontal advection (ΔIPR of 10.2 µg $m^{-3}$ $h^{-1}$). In general, in the growth stage of
haze episodes, the feedback increased the accumulation rate of aerosols mainly through
enhancing chemical formations, weakening vertical diffusions and/or enhancing regional
transport by advections. The feedback-induced changes in the change rate of $PM_{2.5}$
concentration were small during the persistence stage, and the feedback enhanced the
removal rate of $PM_{2.5}$ in the dissipation stage mainly through increasing vertical diffusion
and/or vertical advection.

The results from this study demonstrated a significant impact of aerosol radiative

feedback on meteorology, chemistry, aerosol distribution and evolution during winter haze
events in the BTH region. The mechanism and processes through which the feedback affected
haze formation and evolution were elucidated and quantified. More cases in different regions,
seasons and years are needed to investigate the feedback mechanism at different scales and in
more details. This study also pointed out the significance and necessity of developing online
coupled model for exploring chemistry/aerosol-weather/climate interactions and for





improving meteorological and chemical predictions in both air quality and climate research in
the future.

**Author Contributions**
ZH designed the study, JL performed the model simulation, JL and ZH processed and
analyzed the modeling data, ZH and JL wrote the paper, JL and ZX contributed to the model
development, YW provided and analyzed the chemical observation data, XX provided the
meteorological sounding and aerosol optical observation data, JL and LL processed and
analyzed the observational data, RZ synthesized and analyzed the observation.

**Data availability.**
The observational data can be accessed through contacting the corresponding authors.

**Competing interests.**
The authors declare that they have no conflict of interests.

**Special issue statement.**
This article is part of the special issue "Regional assessment of air pollution and climate
change over East and Southeast Asia: results from MICS-Asia Phase III". It is not associated
with a conference.

**Acknowledgement.**
This study was supported by the National Natural Science Foundation of China (no.
91644217), the National Key R&D Program of China (2019YFA0606802) and the Jiangsu
Collaborative Innovation Center for Climate Change.

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

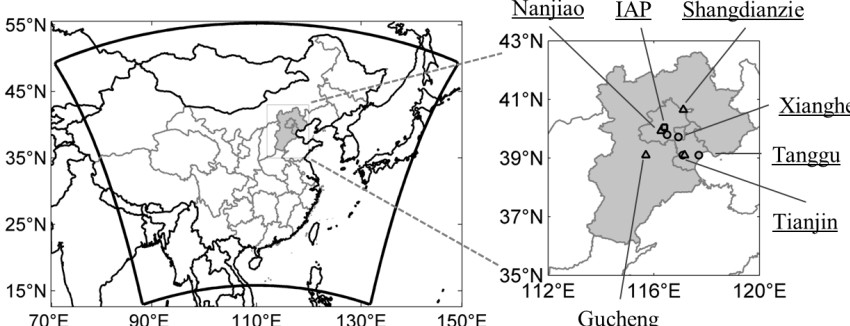

Figure 1. The model study domain. The shaded areas indicate the Beijing-Tianjin-Hebei (BTH) region. Markers are observation sites (square: IAP, observations of PM$_{2.5}$, its chemical components, aerosol extinction coefficient (EXT) and aerosol absorption coefficient (ABS); circles: observations of meteorological variables; triangles: aerosol optical depth. The Xianghe site provides meteorological soundings and hourly surface shortwave radiation (SWDOWN) measurements, the Tianjin site provides both meteorological variables and AOD).



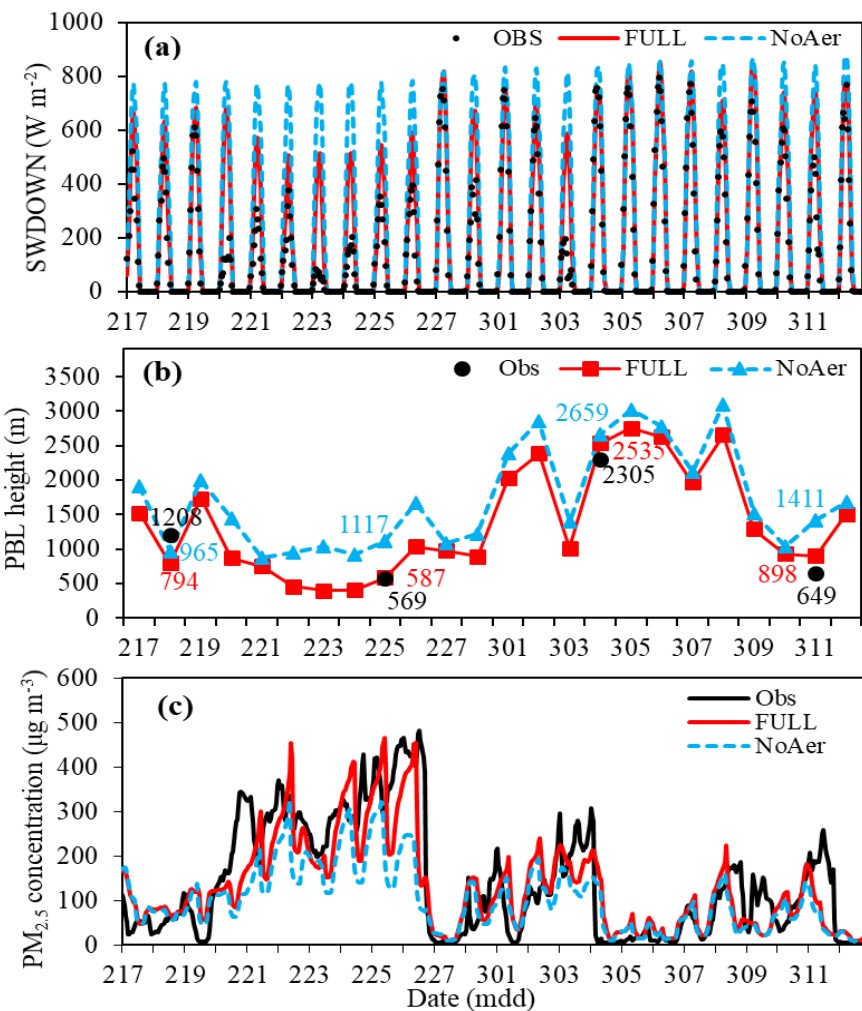

Figure 2. The model simulated and observed (a) hourly SWDOWN at Xianghe, (b) hourly PBL height at 14:00 (LST) at Xianghe (note observations are available in the 4 days, numbers are observations and corresponding simulations) and (c) hourly PM$_{2.5}$ concentration at IAP in Beijing.

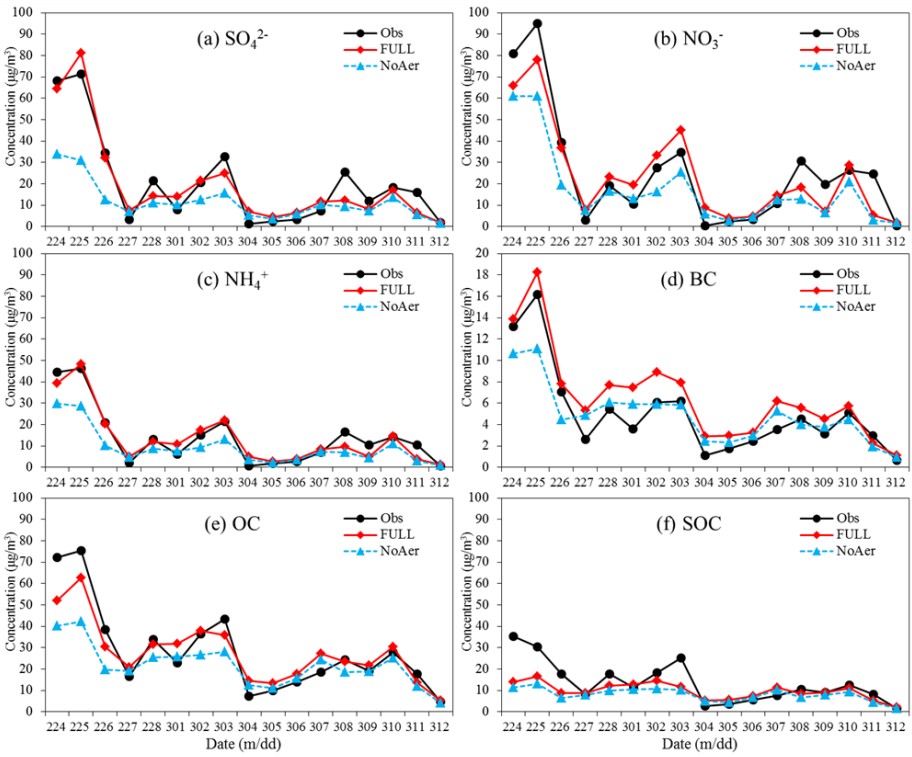


Figure 3. The model simulated and observed daily mean concentrations of aerosol
compositions in PM$_{2.5}$ at the IAP site in Beijing.



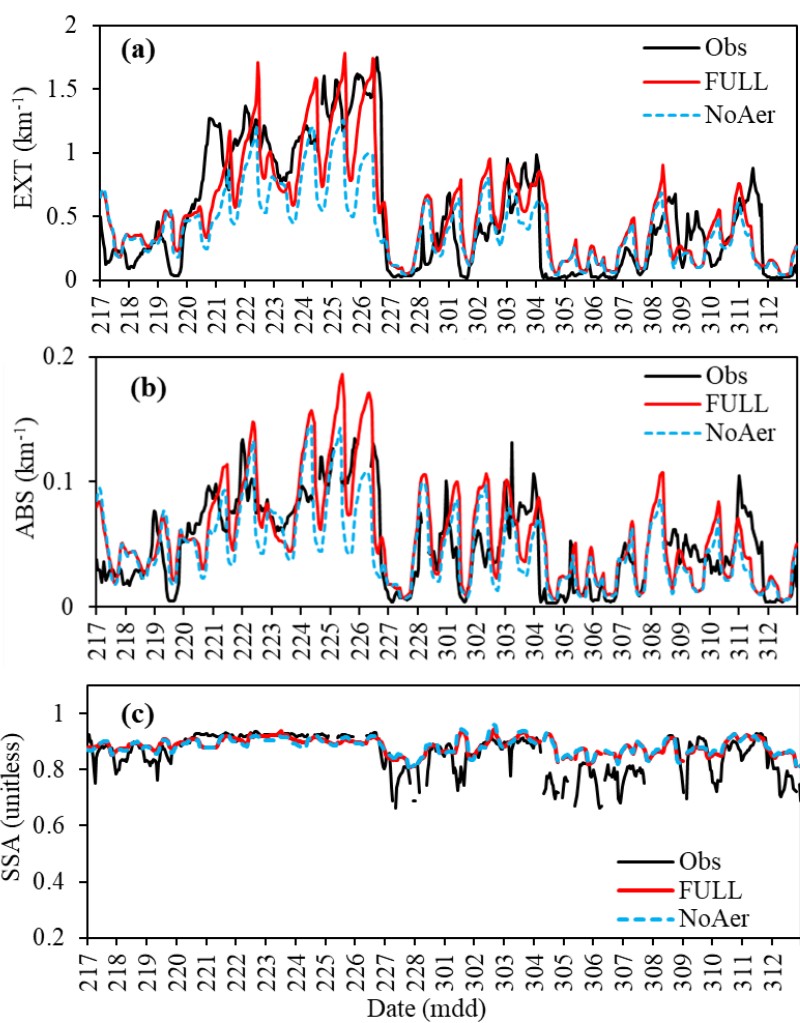


Figure 4. The model simulated and observed hourly (a) aerosol extinction coefficient (EXT),
(b) absorption coefficient (ABS) and (c) single scattering albedo (SSA) at the IAP site in
Beijing under dry condition (RH=10%).



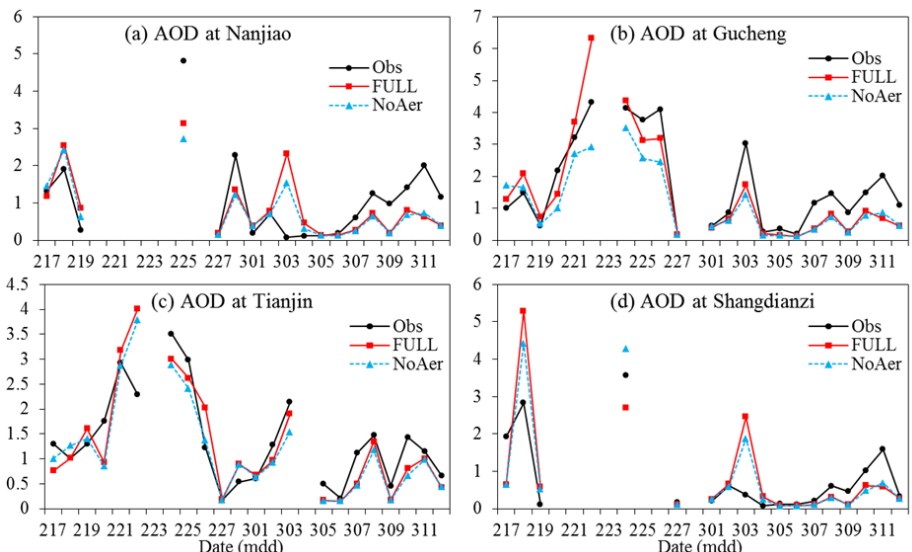

Figure 5. The model simulated and observed daily mean AOD (at 550 nm) at the four sites of CARSNET.

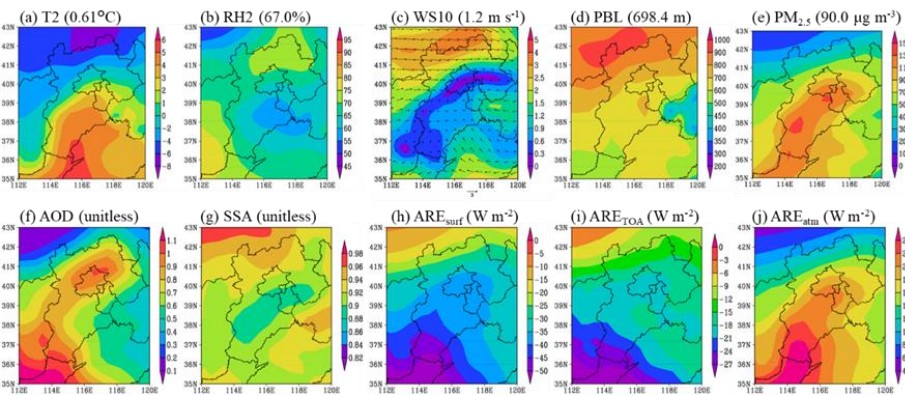

Figure 6. The model simulated (a) air temperature (T2), (b) relative humidity (RH2), (c) wind speed (WS10), (d) PBL height, (e) $PM_{2.5}$ concentration, (f) AOD, (g) SSA, (h) all-sky ARE at the surface, (i) all-sky ARE at the top of atmosphere and (j) all-sky ARE in the atmosphere from the FULL case. Numbers in the parentheses are averages over the BTH region during the entire study period.

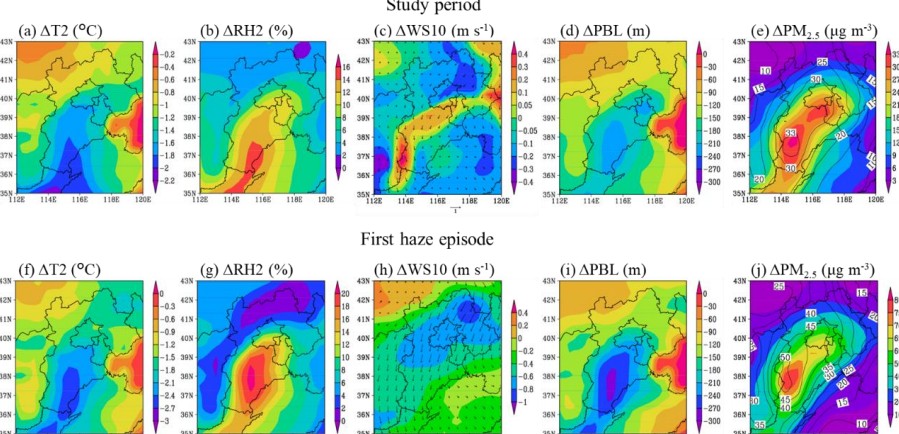

Figure 7. The model simulated feedback-induced changes (FULL minus NoAer) in (a, f)    air temperature (T2), (b, g) relative humidity (RH2), (c, h) wind speed (WS10), (d, i) PBL height and (e, j) PM$_{2.5}$ concentration averaged over the entire study period (a-e) and over the first haze episode    (20-26 February) (f-j). Units are given in the parentheses.

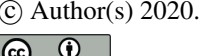

Figure 8. The model calculated integrated process rates (IPR) for the first haze episode (20-26 February) in Beijing. (a) hourly IPR, daily PM$_{2.5}$ concentration and the division of the four stages. The constant IPRs of emissions are not shown for clarity. The mean IPRs for (b) PM$_{2.5}$, (c) BC, (d) sulfate (SO$_4^{2-}$), nitrate (NO$_3^-$), and (f) mean meteorological variables in the four stages. Note that zero IPR





values are not listed. Units of T2, RH2, WS10, PBL and $K_z$ are °C, %, m s$^{-1}$, m and m$^2$ s$^{-1}$, respectively.

Figure 9. Same as Figure 8 but for the second haze episode (1-4 March).



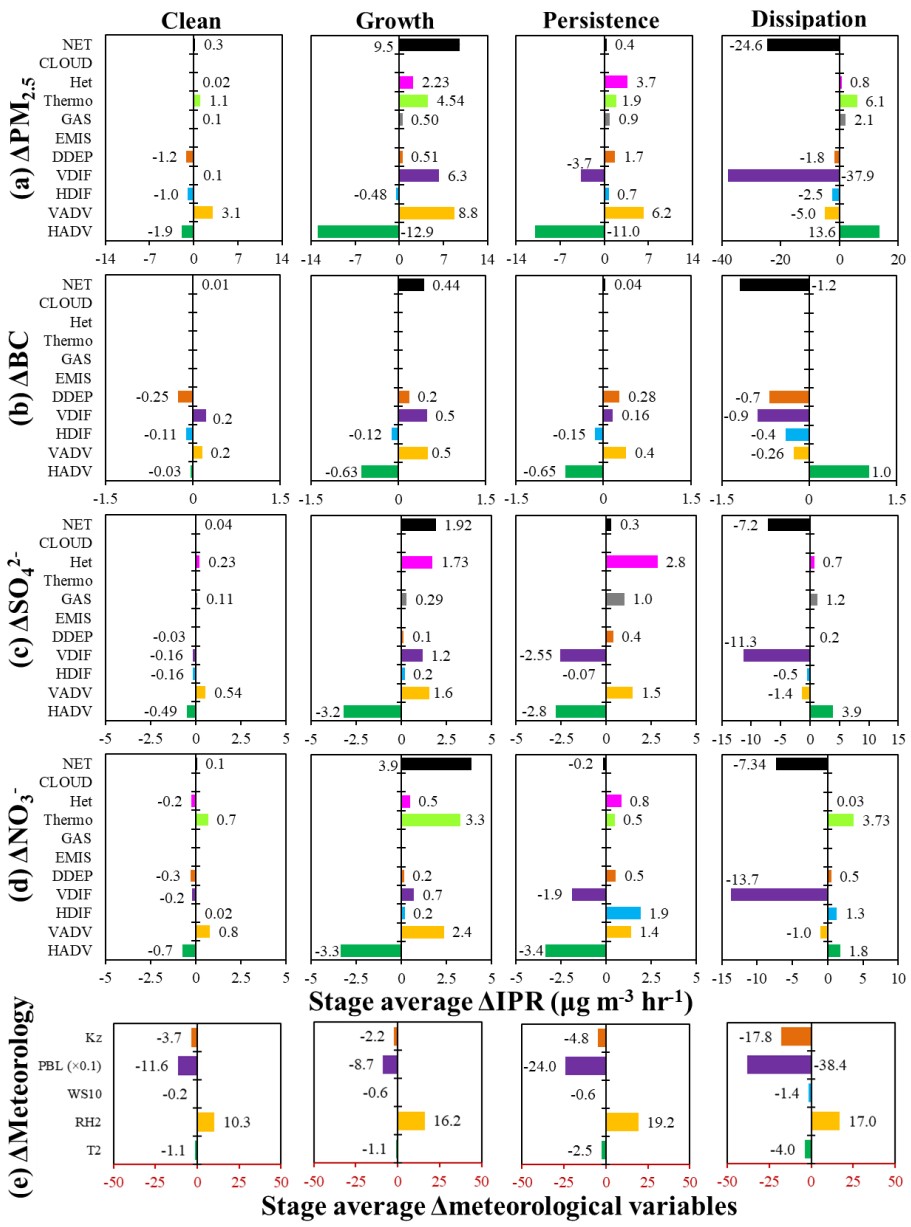

Figure 10. The feedback-induced mean changes in IPRs (FULL minus NoAer) for $PM_{2.5}$ and its chemical components and meteorological variables during the first haze episode (20-26 February) in Beijing. ΔIPRs for $PM_{2.5}$ and its chemical components and Δmeteorological variables are averages over the four stages. Note that zero ΔIPR values (no change) are not shown and the ΔPBL heights are scaled by 0.1. The division of the four stages and units are the same as those in Figure 8.

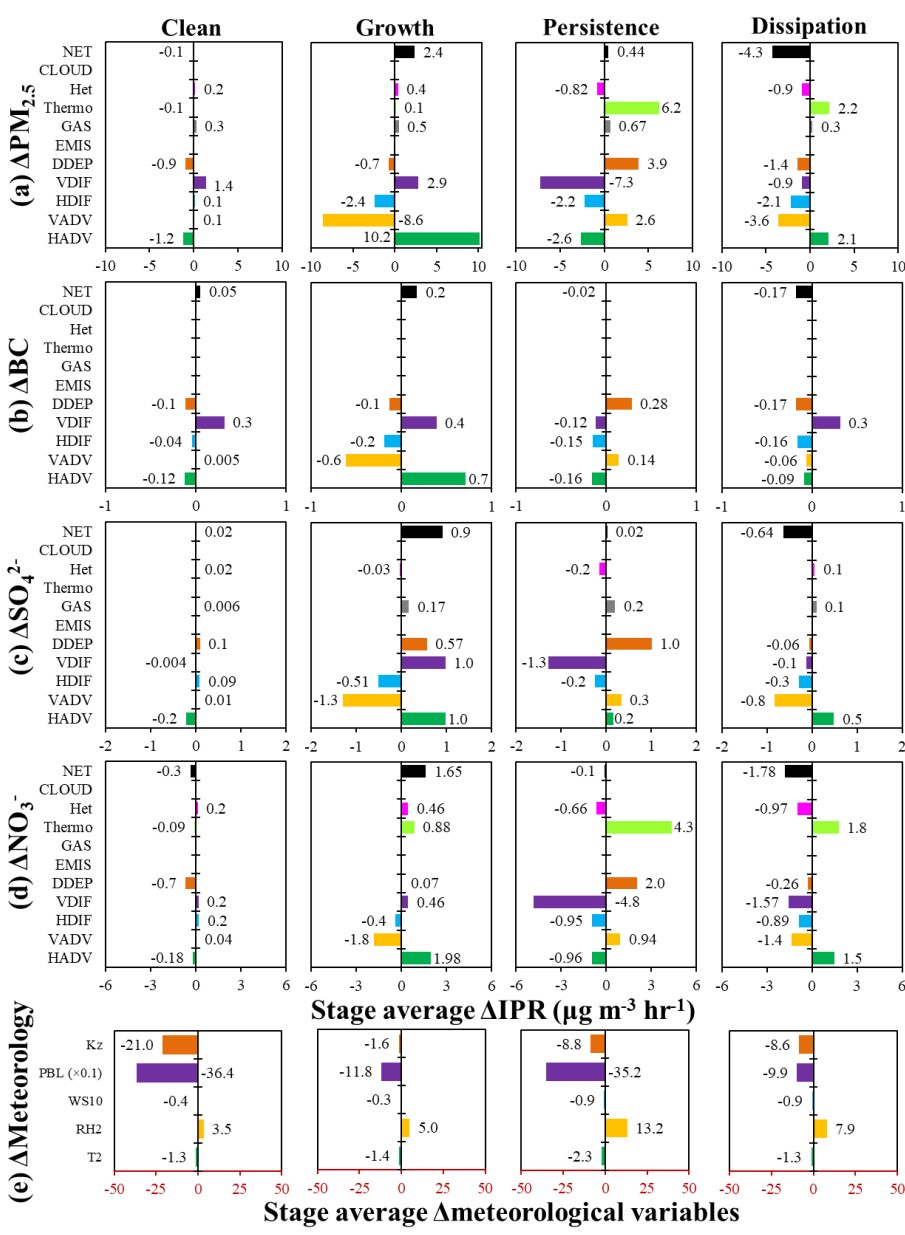

Figure 11. Same as Figure 10 but for the second haze episode (1-4 March). The division of the four stages are the same as that in Figure 9.





Table 1. Performance statistics for meteorological variables at observation sites in the BTH region. Mean observation (Obs), mean simulation (Sim), correlation coefficient (R) and normalized mean bias (NMB in %) are given. WS10, T2 and RH2 are wind speed at 10 meter, air temperature at 2 meter and relative humidity at 2 meter, respectively. All the sample numbers are 207.

| Sites | Longitude | Latitude | WS10 (m s$^{-1}$) | | | | T2 (°C) | | | | RH2 (%) | | | | SWDOWN (W m$^{-2}$) | | | |
|---|---|---|---|---|---|---|---|---|---|---|---|---|---|---|---|---|---|---|
| | | | Obs | Sim | R | NMB | Obs | Sim | R | NMB | Obs | Sim | R | NMB | Obs | Sim | R | NMB |
| **FULL** | | | | | | | | | | | | | | | | | | |
| Beijing | 39°48'N | 116°30'E | 2.3 | 2.9 | 0.53 | 28% | 3.0 | 2.5 | 0.77 | -16% | 53.4 | 62.6 | 0.72 | 17% | | | | |
| Tianjin | 39°6'N | 117°6'E | 2.6 | 3.1 | 0.53 | 23% | 3.5 | 3.8 | 0.89 | 8% | 62.9 | 59.2 | 0.68 | -6% | | | | |
| Tanggu | 39°6'N | 117°42'E | 2.4 | 3.4 | 0.36 | 42% | 3.0 | 3.1 | 0.84 | 2% | 69.3 | 61.3 | 0.49 | -12% | | | | |
| Total | | | 2.4 | 3.1 | 0.47 | 31% | 3.2 | 3.1 | 0.83 | -2% | 61.9 | 61.0 | 0.61 | -1% | | | | |
| Xianghe | 39°45'N | 116°58'E | | | | | | | | | | | | | 136.0 | 188.4 | 0.91 | 38% |
| **NoAer** | | | | | | | | | | | | | | | | | | |
| Beijing | 39°48'N | 116°30'E | 2.3 | 3.4 | 0.48 | 48% | 3.0 | 4.1 | 0.74 | 37% | 53.4 | 51.1 | 0.68 | -4% | | | | |
| Tianjin | 39°6'N | 117°6'E | 2.6 | 3.6 | 0.48 | 39% | 3.5 | 5.3 | 0.88 | 51% | 62.9 | 47.8 | 0.65 | -24% | | | | |
| Tanggu | 39°6'N | 117°42'E | 2.4 | 3.8 | 0.28 | 60% | 3.0 | 4.5 | 0.84 | 50% | 69.3 | 51.4 | 0.48 | -26% | | | | |
| Total | | | 2.4 | 3.6 | 0.41 | 49% | 3.2 | 4.6 | 0.82 | 46% | 61.9 | 50.1 | 0.59 | -19% | | | | |
| Xianghe | 39°45'N | 116°58'E | | | | | | | | | | | | | 136.0 | 234.0 | 0.85 | 72% |





Table 2. Performance statistics for PM$_{2.5}$ concentration and its chemical components, aerosol optical parameters at RH=10% (EXT, ABS and SSA) at the IAP site in Beijing. Mean observation (Obs), mean simulation (Sim), correlation coefficient (R) and normalized mean bias (NMB in %) are listed.

| Species (unit) | Samples | Obs | FULL Sim | R | NMB | NoAer Sim | R | NMB |
|---|---|---|---|---|---|---|---|---|
| PM$_{2.5}$ (µg m$^{-3}$) | 570 | 142.0 | 131.4 | 0.80 | -7% | 101.2 | 0.73 | -29% |
| SO$_4^{2-}$ (µg m$^{-3}$) | 33 | 21.0 | 20.3 | 0.92 | -4% | 11.9 | 0.88 | -44% |
| NO$_3^-$ (µg m$^{-3}$) | 33 | 26.0 | 24.3 | 0.88 | -6% | 17.6 | 0.87 | -32% |
| NH$_4^+$ (µg m$^{-3}$) | 33 | 14.1 | 13.9 | 0.91 | -2% | 9.4 | 0.89 | -34% |
| BC (µg m$^{-3}$) | 33 | 5.2 | 6.7 | 0.92 | 28% | 5.0 | 0.84 | -3% |
| OC (µg m$^{-3}$) | 33 | 29.1 | 28.3 | 0.88 | -3% | 22.3 | 0.78 | -24% |
| POC (µg m$^{-3}$) | 33 | 15.5 | 18.4 | 0.93 | 19% | 14.1 | 0.87 | -9% |
| SOC (µg m$^{-3}$) | 33 | 13.6 | 9.9 | 0.56 | -27% | 8.2 | 0.45 | -40% |
| EXT (km$^{-1}$) | 570 | 0.51 | 0.53 | 0.79 | 4% | 0.41 | 0.72 | -19% |
| ABS (km$^{-1}$) | 534 | 0.048 | 0.052 | 0.68 | 10% | 0.043 | 0.59 | -11% |
| SSA (unitless) | 534 | 0.85 | 0.88 | 0.65 | 5% | 0.88 | 0.59 | 5% |




Table 3. Performance statistics for daily mean AOD at the four CARSNET sites in the BTH region. Mean observation (Obs), mean simulation (Sim), correlation coefficient (R) and normalized mean bias (NMB, in the unit of %) are listed.

|  | Samples | Obs | FULL | | | NoAer | | |
|---|---|---|---|---|---|---|---|---|
|  |  |  | Sim | R | NMB | Sim | R | NMB |
| Nanjiao | 18 | 1.09 | 0.92 | 0.67 | -15.6% | 0.82 | 0.74 | -24.9% |
| Gucheng | 22 | 1.73 | 1.51 | 0.90 | -12.8% | 1.16 | 0.91 | -33.0% |
| Tianjin | 22 | 1.37 | 1.29 | 0.86 | -5.7% | 1.19 | 0.86 | -12.8% |
| Shangdianzi | 17 | 0.84 | 0.90 | 0.72 | 6.2% | 0.89 | 0.85 | 5.0% |
| Total | 79 | 1.29 | 1.18 | 0.81 | -8.6% | 1.03 | 0.82 | -20.2% |

Table 4. The model simulated domain and period averages of AOD, SSA and AREs from the FULL case over the BTH region.

|  | AOD (unitless) | SSA (unitless) | $ARE_{surf}$ $(W\,m^{-2})$ | $ARE_{TOA}$ $(W\,m^{-2})$ | $ARE_{atm}$ $(W\,m^{-2})$ |
|---|---|---|---|---|---|
|  | Study period (Feb 17 - Mar 12) | | | | |
| All day | 0.78 | 0.91 | -37 | -18 | 19 |
| Daytime | 1.53 | 0.92 | -79 | -39 | 40 |
|  | Haze episode 1 (Feb 20-26) | | | | |
| All day | 1.59 | 0.93 | -57 | -32 | 25 |
| Daytime | 3.17 | 0.93 | -123 | -69 | 53 |



Table 5. The model simulated feedback-induced changes (FULL minus NoAer) in T2, WS10, RH2, PBL height, $PM_{2.5}$ concentration and vertical diffusion coefficient ($K_z$) averaged over the BTH region during the entire period and the first haze episode. Inside the parentheses are percentage changes relative to the NoAer case.

| | $\Delta$T2 (°C) | $\Delta$WS10 (m s$^{-1}$) | $\Delta$RH2 (%) | $\Delta$PBL height (m) | $\Delta PM_{2.5}$ ($\mu$g m$^{-3}$) | $\Delta K_z$ (m$^2$ s$^{-1}$) |
|---|---|---|---|---|---|---|
| | | | Study period (Feb 17 - Mar 12) | | | |
| All day | -1.4 (-69.4%) | -0.038 (-3.1%) | +8.7 (+14.9%) | -160.0 (-18.6%) | +20.0 (+28.6%) | -3.3 (-27.0%) |
| Daytime | -1.8 (-42.1%) | +0.028 (+1.9%) | +9.0 (+16.9%) | -267.1 (-22.4%) | +21.1 (+35.6%) | -6.7 (-27.6%) |
| | | | First haze episode (Feb 20-26) | | | |
| All day | -1.8 (-59.7%) | -0.52 (-19.5%) | +9.8 (+12.4%) | -183.6 (-31.0%) | +45.1 (+38.7%) | -3.9 (-48.8%) |
| Daytime | -2.5 (-46.6%) | -0.59 (-19.8%) | +10.4 (+13.8%) | -307.3 (-37.6%) | +49.3 (+48.5%) | -8.3 (-51.9%) |

Table 6. Same as Table 5 but for Beijing.

| | $\Delta$T2 (°C) | $\Delta$WS10 (m s$^{-1}$) | $\Delta$RH2 (%) | $\Delta$PBLH (m) | $\Delta PM_{2.5}$ ($\mu$g m$^{-3}$) | $\Delta K_z$ (m$^2$ s$^{-1}$) |
|---|---|---|---|---|---|---|
| | | | Study period (Feb 17 - Mar 12) | | | |
| All day | -1.6 (-39.1%) | -0.48 (-13.9%) | +11.8 (+23.3%) | -154.0 (-18.3%) | +30.1 (+29.8%) | -4.5 (-37.5%) |
| Daytime | -2.3 (-33.1%) | -0.52 (-13.9%) | +12.5 (+28.1%) | -282.7 (-22.5%) | +34.0 (+43.9%) | -9.6 (-38.8%) |
| | | | First haze episode (Feb 20-26) | | | |
| All day | -2.1 (-46.1%) | -0.58 (-20.4%) | +17.0 (+24.5%) | -195.6 (-35.9%) | +68.0 (+39.1%) | -5.0 (-59.5%) |
| Daytime | -3.4 (-44.6%) | -0.78 (-23.9%) | +17.9 (+27.2%) | -358.3 (-45.5%) | +83.2 (+59.6%) | -11.0 (-63.2%) |



Table 7. The model simulated feedback-induced changes (FULL minus NoAer) in BC, sulfate ($SO_4^{2-}$) and nitrate ($NO_3^-$) averaged over the BTH region and Beijing during the entire period and the first haze episode. Inside the parentheses are percentage changes relative to the NoAer case.

| | Beijing-Tianjin-Hebei region (BTH) | | | Beijing | | |
|---|---|---|---|---|---|---|
| | $\Delta BC$ ($\mu g\ m^{-3}$) | $\Delta SO_4^{2-}$ ($\mu g\ m^{-3}$) | $\Delta NO_3^-$ ($\mu g\ m^{-3}$) | $\Delta BC$ ($\mu g\ m^{-3}$) | $\Delta SO_4^{2-}$ ($\mu g\ m^{-3}$) | $\Delta NO_3^-$ ($\mu g\ m^{-3}$) |
| | Study period (Feb 17 – Mar 12) | | | | | |
| All day | +0.9 (+25.1%) | +5.0 (+46.4%) | +6.8 (+37.3%) | +1.6 (+27.5%) | +8.4 (+58.5%) | +8.4 (+36.9%) |
| Daytime | +1.0 (+39.5%) | +5.4 (+60.2%) | +7.2 (+43.2%) | +1.9 (+51.5%) | +9.5 (+86.5%) | +9.5 (+48.8%) |
| | First haze episode (Feb 20-26) | | | | | |
| All day | +1.9 (+32.9%) | +12.6 (+66.9%) | +14.6 (+40.9%) | +3.1 (+33.6%) | +22.3 (+81.8%) | +16.7 (+34.7%) |
| Daytime | +2.2 (+50.1%) | +13.8 (+81.4%) | +15.8 (+48.3%) | +4.1 (+62.3%) | +26.0 (+112.4%) | +20.9 (+51.5%) |