# Peer review of "Aerosol radiative effects and feedbacks on boundary layer meteorology and 1 PM2.5 chemical components during winter haze events over the 2 Beijing-Tianjin-Hebei region 3 4 Jiawei Li1, Zhiwei Han\*1,2, Yunfei Wu1, Zhe Xiong1, Xiangao Xia"

_Atmospheric Chemistry and Physics, 2020_

## Referee Comment (RC1) · Anonymous Referee #1 · 7 Apr 2020

Aerosol radiative effect (or so called aerosol-radiation interaction, ARI) has been demonstrated to play an important role in pollution deterioration near surface, especially during hazy days. This work developed an online coupled regional chemistry climate model to investigate the mechanisms of ARI on haze pollution. It was demonstrated that there existed a significant impact of aerosol radiative feedback on meteorology, chemistry, aerosol distribution and evolution during winter haze events. One strength of this paper is that the numerical simulation was evaluated against comprehensive observational datasets, like meteorological fields, mass concentration of

multiple pollutants as well as aerosol optical properties. Overall, this work is well structured and written but still needs more in-depth analysis to further improve this article. It worths being published in ACP after addressing the following issues.

Since the work mainly focuses on the impact of aerosol radiative effects on meteorology and the subsequent haze pollution, the model descriptions in Section2 ought to provide more detailed information on how aerosols' optical properties are treated in the model and the method of the online coupling with physical parts.

In terms of the model configurations, the spatial resolution of the model seems a little coarse to characterize aerosol radiative effect on the atmospheric stratification, especially in BTH region with complex terrain. There were 16 vertical layers in the vertical dimension, as described in Section 2. How were these vertical grids distributed in the simulations? As demonstrated in previous related works (Wilcox et al., 2016;Wang et al., 2018;Huang et al., 2018), both temperature stratification and aerosol vertical profile, which are vital for aerosol's impact on near-surface pollution, are very sensitive to the vertical grid settings in models. Given that aerosol radiative effect features surface cooling and atmospheric warming and thus more stable stratification, insufficiently fine resolution may partly offset these two opposite tendency and underestimate the pollution deterioration.

Minor concerns:

Line 310-312: It is a little confusing about the definition of the NoAer simulation. Did it include aerosol-cloud interaction? Or it excluded any impact of aerosol on meteorology? Line 856-864: the thermodynamic process of nitrate aerosol is also highly dependent on the air temperature. As shown in Fig. 7, in addition to RH increase, 2-meter temperature decrease significantly and may contribute to the gas-aerosol partitioning and subsequent nitrate formation.

Another minor issue is that most of the labels in all the figures, including the coordinate axis, are too small to be clearly identified. It needs to be improved in the revision.

[Figure]

References: Wang, Z., et al.,: Dome effect of black carbon and its key influencing factors: a one-dimensional modelling study, Atmos. Chem. Phys., 18, 2821–2834, https://doi.org/10.5194/acp-18-2821-2018, 2018.

Huang, X., et al.,: Impact of Aerosol-PBL Interaction on Haze Pollution: Multiyear Observational Evidences in North China, Geophys. Res. Lett., 45, 8596-8603, 10.1029/2018gl079239, 2018

Wilcox, E. M., Thomas, R. M., Praveen, P. S., Pistone, K., Bender, F. A., & Ramanathan, V. (2016). Black carbon solar absorption suppresses turbulence in the atmospheric boundary layer. Proceedings of the National Academy of Sciences of the United States of America, 113( 42), 11,794– 11,799.
* * *

---

## Referee Comment (RC2) · Anonymous Referee #2 · 11 May 2020

The manuscript investigated the aerosol radiative effects (AREs) on meteorology and particulate matter (PM) pollution in Bejing-Tianjin-Hebei (BTH) using a fully coupled chemistry transport model. The topic is of interest and within the scope of ACP. However, there are several factors hindering publication of the manuscript at the present form.

General comments:

1) Two-page abstract is lengthy and tedious. The authors need to abbreviate it and

make it concise. I suggest that the authors put more emphases on results of process analyses. In addition, the domain average is the BTH average? I do not believe that domain average PM2.5 enhancement due to AREs is 29%, which is rather large considering that the domain covers the East Asia.

2) AREs include aerosol direct scattering and/or absorbing of incoming solar radiation and the induced adjustments of the surface energy budget, thermodynamic profile and cloudiness (direct effects and semi-direct effects), and serving as cloud condensation nuclei and ice nuclei to alter cloud properties such as cloud lifetime, reflectivity and composition (indirect effects). IPCC (2013) has use the new terminology of the aerosol-radiation interactions (ARI) to represent the combination of the aerosol direct and semi-direct effects, and use the aerosol-cloud interactions (ACI) to represent the aerosol indirect radiative effects. I am not sure whether the authors considered both ARI and ACI. Lines 309-312, they clarified that they shut off all AREs, i.e., ARI and ACI. However, ACI could not be shut off! Therefore, the authors need to clarify whether ACI were included or not. If so, which microphysical scheme is used and how to consider aerosol activation to cloud condensation nuclei and ice nuclei.

3) Model validation. Air pollutants measurement has been released since 2013 in BTH by China Ministry of Ecology and Environment. The authors only compared PM2.5 simulations with measurement at IAP, Beijing, which is not sufficient. Temporal and spatial validations of PM2.5, O3, NO2, and SO2 in BTH are necessary to warrant reasonable simulations, particularly for O3, NO2, and SO2 which are the key to sulfate and nitrate simulations. Furthermore, I am rather surprised that the authors used a two-product model to yield high SOA levels.

Specific comments:

Lines 72-73, please provide references to support your clarification.

Lines 78-79, "east China" include "north China" or vice versa?

[Figure]

Line 80, "haze pollution" is a little bit odd. Haze is a weather phenomenon with a horizontal visibility of less than 10 km, which might be caused by high levels of PM. Please change "haze pollution" to "PM pollution during haze days" throughout the manuscript.

Lines 88-90, please provide appropriate references.

Line 93, I do not think that Grell et al. (2005) have studied AREs on air quality and meteorology.

Line 106, "however" should read "but".

Lines 125-128, I fairly disagree with the authors' clarification about simulations of secondary aerosols. There still exist large gaps for SOA between simulations and observations, but models can generally well produce sulfate and nitrate.

Lines 167-169, please provide references.

Lines 167-180, which microphysical scheme is used in RIEMS-Chem? Does it consider ACI?

Lines 181-196, Does RIEMS-Chem include aerosol effects on photolysis?

Lines 197-208, how does RIEMS-Chem simulate the nitrate formation? Is the organic coating considered in calculation of N2O5 hydrolysis?

Lines 209-226, the authors clarified that they used different size distribution for inorganic, black and organic carbon aerosols. However, they also assumed completely internal mixing aerosols, which should be represented by the same size distribution. In addition, the aerosol size distribution and mixing state change considerably with development of PM pollution. The authors need at least to include discussions on uncertainties in ARE caused by variations of aerosol size distribution and mixing state.

Lines 227-231, the authors have used ISORROPIA to calculate aerosol water, so it might not be appropriate to consider the hygroscopic growth of inorganic aerosols, BC, POA and SOA.

Lines 293-298, the 60km horizontal resolution is too coarse to focus on the Beijing metropolitan.

---

## Author Comment (AC1) · 19 May 2020

The comment was uploaded in the form of a supplement

———————————

---

## Author Comment (AC2) · 19 May 2020

The comment was uploaded in the form of a supplement
* * *

---

## Author Comment (AC3) · 19 May 2020

**Responses to the reviewer's comments**

MS No.: acp-2020-182
Title: Aerosol radiative effects and feedbacks on boundary layer meteorology and PM2.5 chemical components during winter haze events over the Beijing-Tianjin-Hebei region

The authors greatly appreciate the valuable and constructive comments from the two reviewers, which have helped us improve the manuscript. We have responded to their comments carefully and revised the manuscript accordingly by taken their good suggestions into account. The detailed responses (blue font) are as follows:

*Response to Referee #1*

*General comments:*
Aerosol radiative effect (or so called aerosol-radiation interaction, ARI) has been demonstrated to play an important role in pollution deterioration near surface, especially during hazy days. This work developed an online coupled regional chemistry climate model to investigate the mechanisms of ARI on haze pollution. It was demonstrated that there existed a significant impact of aerosol radiative feedback on meteorology, chemistry, aerosol distribution and evolution during winter haze events. One strength of this paper is that the numerical simulation was evaluated against comprehensive observational datasets, like meteorological fields, mass concentration of multiple pollutants as well as aerosol optical properties. Overall, this work is well structured and written but still needs more in-depth analysis to further improve this article. It worths being published in ACP after addressing the following issues.

**Reply:** Thanks for the valuable and constructive comments which help us improve the manuscript. We have responded to your comments in detail and revised the manuscript as your suggestions.

*Specific comments:*
1. Since the work mainly focuses on the impact of aerosol radiative effects on meteorology and the subsequent haze pollution, the model descriptions in Section 2 ought to provide more detailed information on how aerosols' optical properties are treated in the model and the method of the online coupling with physical parts.

**Reply:** Thanks for the good suggestion. We add more detailed information on the treatment and calculation of optical properties and the coupling method between chemical and physical parts in the revised version as follows:
"Mass concentrations of aerosol components are firstly calculated by chemical module. Aerosol number concentration is calculated based on mass concentration and size distribution derived from in-situ observations in Beijing. Then all the information including mass concentrations, size distributions, refractive indices for each aerosol component (based on the OPAC dataset), hygroscopicity ($\kappa$) for each component, and ambient meteorological variables are provided to the optical module which is based on the scheme of Ghan and Zaveri (2007)

and calculates the aerosol optical properties (scattering coefficient, extinction coefficient and asymmetry factor). In this scheme, the optical properties of different types of aerosols are pre-calculated by Mie theory and fitted by Chebyshev polynomials, which are functions of aerosol geometric mean diameter and refractive index:

$$Q = \exp\left[\sum_{k=1}^{10} A_k T_k(x)\right], \tag{1}$$

$$x = \frac{2\log(D_p) - \log(D_{min}) - \log(D_{max})}{\log(D_{max}) - \log(D_{min})}, \tag{2}$$

where Q represents the aerosol optical properties (such as scattering efficiency). $T_k(x)$ are the Chebyshev polynomial of order k, $A_k$ are the Chebyshev coefficients, $D_p$ is the geometric mean diameter, $D_{min}$ and $D_{max}$ are the minimum and maximum $D_p$ for obtaining the Chebyshev polynomials, with values of 0.001 μm and 10 μm, respectively. It has been proved that 40 groups of $D_p$ in the range from $D_{min}$ and $D_{max}$ are sufficient to control errors below 10% compared with classical Mie code calculation.

The effect of water uptake is treated by the κ-Köhler parameterization (Petters and Kreidenweis, 2007), which calculates aerosol wet diameter due to hygroscopic growth under different relative humidity. The bulk κ for internal mixture of aerosols is derived by the volume-weighted average of κ of each aerosol component:

$$\kappa = \sum_j \frac{V_j}{V_a} \kappa_j, \tag{3}$$

where $V_a$ is the total volume of dry aerosols, $V_j$ is the volume of each aerosol component j.

The refractive index of internally mixed aerosols is calculated using the Maxwell-Garnett mixing rule:

$$R_w^2 = R_s^2 \left[\frac{R_i^2 + 2R_s^2 + 2f_i(R_i^2 - R_s^2)}{R_i^2 + 2R_s^2 - f_i(R_i^2 - R_s^2)}\right], \tag{4}$$

$$f_i = \frac{V_i}{V}, \tag{5}$$

where $R_w$ is the refractive index of the internal mixture, $R_i$ and $R_s$ are the refractive index of insoluble components (BC and POA) and soluble components (inorganic aerosols, SOA and water), respectively. $V_i$ represents the volume of insoluble components, V represents the total volume of wetted aerosols. In the model, the κ values for inorganic aerosols, BC, POA, SOA, dust and sea salt are set to 0.65, 0, 0.1, 0.2, 0.01 and 0.98, respectively, according to previous observational and modeling studies (Riemer et al., 2010; Liu et al., 2010a; Westervelt et al., 2012)

After obtaining the wet diameter ($D_p$) and refractive index of the internally mixed aerosols ($R_w$), the aerosol optical properties (Q) can be derived from formula (1) with the Chebyshev fitting coefficients table. Then, aerosol optical parameters, such as extinction coefficient can be obtained through multiplying Q by aerosol mass concentration from chemical module. The advantage of this optical module is the computational speed is much faster than that from traditional Mie calculation, with a similar level of accuracy. This module

has been successfully used in estimations of aerosol optical properties and direct radiative effects over East Asia (Han et al., 2011a; Li and Han, 2016b; Li et al., 2019b). (note these papers are already listed in the manuscript)

The aerosol optical parameters and $N_c$ by aerosol activation calculated above are transferred into radiation module to account for the perturbation of radiation and atmospheric heating rate due to aerosol direct and indirect effects. The following land surface module and boundary layer module account for the changes in land-air fluxes of heat and moisture, turbulent diffusion coefficients and meteorological variables in the boundary layer in response to the radiation change, and air temperature tendency is calculated in terms of the altered atmospheric heating rate and radiation, which further lead to changes in meteorological variables, and in turn affect physical and chemical processes and concentrations of aerosols and their precursors treated in the chemical module. All the modules are called every 2.5 minutes and the exchange of variables between chemical module and radiation/meteorological modules is made every 30 minutes". Note we also add detailed information on aerosol-cloud interaction in the revised version, see the response to the reviewer #2.

2. In terms of the model configurations, the spatial resolution of the model seems a little coarse to characterize aerosol radiative effect on the atmospheric stratification, especially in BTH region with complex terrain. There were 16 vertical layers in the vertical dimension, as described in Section 2. How were these vertical grids distributed in the simulations? As demonstrated in previous related works (Wilcox et al., 2016; Wang et al., 2018; Huang et al., 2018), both temperature stratification and aerosol vertical profile, which are vital for aerosol's impact on near-surface pollution, are very sensitive to the vertical grid settings in models. Given that aerosol radiative effect features surface cooling and atmospheric warming and thus more stable stratification, insufficiently fine resolution may partly offset these two opposite tendency and underestimate the pollution deterioration.

**Reply:** In RIMES-Chem, 16 vertical layers distribute vertically and unevenly in the terrain-following sigma coordinate. There about 8 layers within the lowest 2 kilometers, with the first model layer being about 30 meter above ground.

This study focuses on the feedback mechanism over the Beijing-Tianjin-Hebei (BTH) region (most of which is plain), which is a regional problem involving a variety of sources, long range transport and chemical transformation, so application of a regional model characterized by model grids of several tens of kilometer in the horizontal and less than twenty layers in the vertical is acceptable considering the large scale of air pollution and the high computational cost in an online coupled model.

Normally, a finer model grid resolution is considered to produce better model results than coarse one, however, this does not always apply for all conditions. Recently, Tao et al. (2020) examined the impact of model grid resolution on meteorology and air pollutant prediction over the North China Plain (NCP) during 2010 using the NASA Unified Weather Research and Forecasting (NU-WRF) model, with horizontal resolutions at 45, 15 and 5 km, respectively. They found that the improvement of air quality modeling was not linear with the resolution increase, the fine resolution did not necessarily predict better results than coarse resolution, which was attributed to the limitation in the resolution of anthropogenic emission inventory (please also see response to the reviewer #2).

We agree that vertical profiles of aerosols and air temperature are crucial to the estimation of aerosol radiative effect. However, so far, the observation of aerosol vertical profile is very limited in north China and it is not available for this study. We have ever looked for CALIPSO retrievals from the website (https://www-calipso.larc.nasa.gov), but unfortunately the quality of CALIPSO data during the haze episodes of this study is very low to compare with. However, the vertical profiles of aerosol extinction coefficients from RIEMS-Chem have ever been comprehensively evaluated against CALIPSO data over east China including Beijing in October 2010, which demonstrates a good ability of this model in reproducing aerosol vertical distributions (Li and Han, 2016b).

To examine the model performance for meteorology in the vertical direction, we collected meteorological sounding data at Beijing observatory from the website of University of Wyoming (http://weather.uwyo.edu/upperair/sounding.html). Figure S1 and S2 present the observed and simulated average vertical profiles of air temperature, wind speed and relative humidity at 08:00 LST and 20:00 LST during the two haze episodes of 20–26 February and 1–4 March 2014 and the corresponding comparison statistics for these variables (Tables S1 and S2) in the troposphere and at altitudes below 3 km (please see supplement file). In general, the model is able to reasonably reproduce the major features of vertical distribution of key meteorological variables, although the model tends to predict higher relative humidity in the middle-upper troposphere, such overpredictions are also seen for the same region in winter in previous studies, such as WRF-Chem simulation (Gao et al., 2016). The statistics indicate that the model simulated vertical distribution of meteorological variables are within an acceptable accuracy range of current meteorological model predictions.

Considering this study focuses on regional haze events in the BTH region and the computational cost, the adoption of current model grid resolution is acceptable, and given the generally good ability of the model in reproducing spatial distribution and temporal variation of meteorological variables, boundary layer height, aerosol compositions and optical properties, the model results from this study would be reasonable and reliable.

We agree with the reviewer that finer model grid may enhance the prediction accuracy of temperature stratification and aerosol vertical profile and coarser vertical grid resolution might mitigate vertical temperature gradient and possibly underestimates surface pollution deterioration. We will use finer model grid resolution along with finer emission inventory and vertical observations of aerosols when available in the future. We add some discussions on this uncertainty in the conclusion of revised version by citing relevant papers.

*Reference*
Gao, M., Carmichael, G. R., Wang, Y., Saide, P. E., Yu, M., Xin, J., Liu, Z., and Wang, Z.: Modeling study of the 2010 regional haze event in the North China Plain, Atmos. Chem. Phys., 16, 1673-1691, 10.5194/acp-16-1673-2016, 2016.

Huang X., Wang, Z. L., Ding, A. J.: Impact of Aerosol-PBL Interaction on Haze Pollution: Multiyear Observational Evidences in North China, Geophys. Res. Lett., 45, 8596-8603, 2018.

Li, J. W. and Han, Z. W.: Aerosol vertical distribution over east China from RIEMS-Chem simulation in comparison with CALIPSO measurements, Atmos. Environ., 143, 177-189, 2016b.

Tao, Z. N., Chin, M., Gao, M., Kucsera, T., Kim, D.-C., Bian, H. S., Kurokawa, J.-I., Wang, Y. S., Liu, Z. R., Carmichael, G. R., Wang, Z. F., and Akimoto, H.: Evaluation of NU-WRF model performance on air quality simulation under various model resolutions – an investigation within the framework of MICS-Asia Phase III. Atmos. Chem. Phys., 20, 2319–2339, 2020.

Wang, Z. L., Huang, X., Ding, A. J.: Dome effect of black carbon and its key influencing factors: a one-dimensional modelling study, Atmos. Chem. Phys., 18, 2821–2834, 2018.

Wilcox, E. M., Thomas, R. M., Praveen, P. S., Pistone, K., Bender, F. A., and Ramanathan, V.: Black carbon solar absorption suppresses turbulence in the atmospheric boundary layer. Proc. Natl. Acad. Sci. USA, 113(42), 11794–11799, 2016.

Minor concerns:

Line 310-312: It is a little confusing about the definition of the NoAer simulation. Did it include aerosol-cloud interaction? Or it excluded any impact of aerosol on meteorology?

**Reply:** The NoAer simulation shuts off aerosol direct radiative effects and removes anthropogenic aerosols in aerosol indirect effects. We describe it more clearly in the revised version.

Line 856-864: the thermodynamic process of nitrate aerosol is also highly dependent on the air temperature. As shown in Fig. 7, in addition to RH increase, 2-meter temperature decrease significantly and may contribute to the gas-aerosol partitioning and subsequent nitrate formation.

**Reply:** Yes. Low temperature favors the condensation of nitric acid and ammonia into nitrate particle. We changed the relevant sentences to "The substantial increase in the contribution of thermodynamic processes to nitrate production was due to the combined effects of the increased level of nitrate precursors ($HNO_3$ and $NH_3$) resulting from weakened diffusivity and the increased RH along with the decreased air temperature, which were favorable for gas to aerosol conversion" in the revised version.

Another minor issue is that most of the labels in all the figures, including the coordinate axis, are too small to be clearly identified. It needs to be improved in the revision.

**Reply:** We have redrawn figures 3 – 11 with larger labels for legibility in the revised version.

**Response to Referee #2**

The manuscript investigated the aerosol radiative effects (AREs) on meteorology and particulate matter (PM) pollution in Bejing-Tianjin-Hebei (BTH) using a fully coupled chemistry transport model. The topic is of interest and within the scope of ACP. However, there are several factors hindering publication of the manuscript at the present form.

**Reply:** We would like to thank the reviewer for the valuable comments, constructive suggestions and careful reading, which have helped us improve this manuscript.

*General comments:*

1) Two-page abstract is lengthy and tedious. The authors need to abbreviate it and make it concise. I suggest that the authors put more emphases on results of process analyses. In addition, the domain average is the BTH average? I do not believe that domain average PM2.5 enhancement due to AREs is 29%, which is rather large considering that the domain covers the East Asia.

**Reply:** Thanks for the suggestion. We abbreviate the abstract accordingly. Yes, the domain average is the BTH average, we rewrite the sentence for clarity.

2) AREs include aerosol direct scattering and/or absorbing of incoming solar radiation and the induced adjustments of the surface energy budget, thermodynamic profile and cloudiness (direct effects and semi-direct effects), and serving as cloud condensation nuclei and ice nuclei to alter cloud properties such as cloud lifetime, reflectivity and composition (indirect effects). IPCC (2013) has use the new terminology of the aerosol-radiation interactions (ARI) to represent the combination of the aerosol direct and semi-direct effects, and use the aerosol-cloud interactions (ACI) to represent the aerosol indirect radiative effects. I am not sure whether the authors considered both ARI and ACI. Lines 309-312, they clarified that they shut off all AREs, i.e., ARI and ACI. However, ACI could not be shut off! Therefore, the authors need to clarify whether ACI were included or not. If so, which microphysical scheme is used and how to consider aerosol activation to cloud condensation nuclei and ice nuclei.

**Reply:** We are sorry for the confusion and missing of description on aerosol indirect effects. We include both ARI and ACI in the model simulation. An empirical method from Hegg (1994) is applied to link cloud droplet number concentration $N_c$ to mass concentration of hydrophilic aerosols (sulfate, nitrate, hydrophilic BC and OC) to represent the first indirect effect, while the parameterization of Beheng (1994) is used to represent the second indirect effect, in which the autoconversion rate converting from cloud water to rain water depends on $N_c$ and cloud liquid water content $W_L$. The cloud effective radius $r_e$ is calculated based on $N_c$, $W_L$ and the cube of the ratio of the mean volume radius and the effective radius of the cloud-droplet spectrum following Martin et al. (1994). ARI is shut off by inactive aerosol feedback to radiation, ACI is closed by zeroing anthropogenic aerosols in the Hegg's scheme, leaving $N_c$ to be a prescribed background value of 23/cm$^3$. So we agree that "shut off ACI" is not appropriate because ACI still works even without anthropogenic aerosols, we change this sentence to "the NoAer simulation shuts off aerosol direct radiative effects and removes anthropogenic aerosols in aerosol indirect effects" in the revised version. The effect of aerosols on ice nuclei and convective cloud is not treated yet in this model because of the complexity and limitation in knowledge. It was noticed that these was relatively less cloud cover in the BTH during the study period, so as we described in the previous version (line 579-581) "The indirect radiative effect was … much smaller than the direct radiative effect; therefore, the total radiative feedback is predominated by direct radiative effect during the study period".

*Reference*
Beheng, K. D.: A parameterization of warm cloud microphysical conversion processes,

Atmospheric Research, 33, 193–206, 1994.

Hegg D. A.: Cloud condensation nucleus-sulfate mass relationship and cloud albedo. J. Geophys. Res.:Atmosphere., 99, D12, 25903-25907, 1994.

Martin, G. M., Johnson, D. W., and Spice, A.: The Measurements and Parameterization of Effective Radius of Droplets in Warm Stratocumulus Clouds, J. Atmos. Sci., 51, 1823–1842, 1994.

3) Model validation. Air pollutants measurement has been released since 2013 in BTH by China Ministry of Ecology and Environment. The authors only compared PM2.5 simulations with measurement at IAP, Beijing, which is not sufficient. Temporal and spatial validations of PM2.5, O3, NO2, and SO2 in BTH are necessary to warrant reasonable simulations, particularly for O3, NO2, and SO2 which are the key to sulfate and nitrate simulations. Furthermore, I am rather surprised that the authors used a two-product model to yield high SOA levels.

**Reply:** Thanks for the good suggestions. We collected observations at 80 surface stations in 13 cities in the BTH region from the website of CNEMC (China National Environmental Monitoring Center) (http://www.cnemc.cn/) and made a detailed comparison between observations and model simulations for $PM_{2.5}$, $O_3$, $NO_2$ and $SO_2$ as your suggestion. The observed and simulated hourly mass concentrations of these species in several typical cities are presented in Figure S3-S8 and the comparison statistics for each city and for all the cities are presented in Table S3 (please see the supplement file). The overall model performance is generally satisfactory, with Rs of 0.87, 0.81, 0.60 and 0.74, NMBs of -0.4%, -11%, -17% and 0.5% for $PM_{2.5}$, $O_3$, $SO_2$ and $NO_2$, respectively, for all the sites in the BTH region. We redraw Figure 1 and add the above comparison in the supplement file in the revised version.

In this model, we use a two-product model to calculate SOA with corrected stoichiometric coefficients for semi-volatile gases, about 4 times the previous ones considering the influence of vapor wall losses (Baker et al., 2015). Actually, the model still underpredicts SOA level by a factor of 2-3 during haze episodes (Figure 3f), which indicates the large bias in SOA simulation by the two-product model. We plan to implement the VBS approach in the model in the future but it needs much computational cost. In terms of observation, the inorganic aerosol concentrations (the sum of sulfate, nitrate and ammonium, 61.1 µg m$^{-3}$) are much larger than SOA concentration (13.6 µg m$^{-3}$), which dominates the $PM_{2.5}$ mass during the study period.

*Reference*

Baker, K. R., Carlton, A. G., Kleindienst, T. E., Offenberg, J. H., Beaver, M. R., Gentner, D. R., Goldstein, A. H., Hayes, P. L., Jimenez, J. L., Gilman, J. B., de Gouw, J. A.,Woody, M. C., Pye, H. O. T., Kelly, J. T., Lewandowski, M., Jaoui, M., Stevens, P. S., Brune, W. H., Lin, Y. H., Rubitschun, C. L., and Surratt, J.D.: Gas and aerosol carbon in California: comparison of measurements and model predictions in Pasadena and Bakersfield, Atmos. Chem. Phys., 15, 5243-5258, 2015.

*Specific comments:*
Lines 72-73, please provide references to support your clarification.

**Reply:** We change the sentences to "The above interactions are traditionally not included or simplified in meteorological or chemical models, but have now been considered and treated with different degrees of complexity in several online coupled models along with the advances in our knowledge and computer power, and the coupling of meteorology and chemistry and its feedbacks remains one of the most challenging issues in air quality and climate change (Zhang, 2008; Baklanov et al., 2014). " in the revised version.

Lines 78-79, "east China" include "north China" or vice versa?

**Reply:** We change "north China" to "the BTH region", to our understanding, "east China" includes "the BTH region".

Line 80, "haze pollution" is a little bit odd. Haze is a weather phenomenon with a horizontal visibility of less than 10 km, which might be caused by high levels of PM. Please change "haze pollution" to "PM pollution during haze days" throughout the manuscript.

**Reply:** Thank you for the suggestion, we change "haze pollution" to "haze, haze problem, haze event" or "PM pollution during haze days" throughout the manuscript as suggestion.

Lines 88-90, please provide appropriate references.

**Reply:** References (Fu and Chen, 2017; Zhong et al., 2018a; An et al., 2019) are provided in the revised version. (these papers have already been included in the manuscript)

*Reference*
An, Z., Huang, R., Zhang, R., Tie, X., Li, G., Cao, J., Zhou, W., Shi, Z., Han, Y., Gu, Z., and Ji, Y.: Severe haze in northern China: A synergy of anthropogenic emissions and atmospheric processes, P. Natl. Acad. Sci. USA, 116,18, 8657–8666, 2019.
Fu, H. B. and Chen, J. M.: Formation, features and controlling strategies of severe haze-fog pollutions in China. Sci. Total. Environ., 578, 121–138, 2017.
Zhong, J., Zhang, X., Dong, Y.,Wang, Y., Liu, C.,Wang, J., Zhang, Y., and Che, H.: Feedback effects of boundary-layer meteorological factors on cumulative explosive growth of $PM_{2.5}$ during winter heavy pollution episodes in Beijing from 2013 to 2016, Atmos. Chem. Phys., 18, 247–258, https://doi.org/10.5194/acp-18-247-2018, 2018a.

Line 93, I do not think that Grell et al. (2005) have studied AREs on air quality and meteorology.

**Reply:** Yes, Grell et al. (2005) described the structure, configuration and evaluation of WRF-Chem, we deleted it in the revised version.

Line 106, "however" should read "but".

**Reply:** revised.

Lines 125-128, I fairly disagree with the authors' clarification about simulations of secondary

aerosols. There still exist large gaps for SOA between simulations and observations, but models can generally well produce sulfate and nitrate.

**Reply:** These sentences could be misleading. The online coupled models might produce sulfate and nitrate generally well in other regions of the world, but their performances in China is relatively poor, especially in the BTH region, which has been recently demonstrated by the modeling works in the Model Inter Comparison Study for Asia (MICS-Asia) project. For example, Gao et al. (2018) reported results for aerosol components in winter in the BTH region from seven online coupled model simulations including WRF-Chem, WRF-CMAQ etc., in which 6 simulations largely underpredict sulfate and nitrate observations. Chen et al. (2019) systematically evaluated model simulations for aerosol components over east China and the western Pacific for the year 2010 from 14 online and offline models, including 9 different versions of WRF-Chem and WRF-CMAQ. The validation showed that 13 model simulations consistently underpredicted sulfate observation by up to 68%, with the ensemble mean being 19% lower than observation, and larger disparity existed in model simulations for nitrate concentration, with the normalized mean biases in a range of -81~125%. The above modeling studies indicates the limitation of current models in reproducing inorganic aerosols. So far, there is still an argument on the dominant mechanism for rapid sulfate formation during haze episodes in the north China Plain (Cheng et al., 2016; Wang et al., 2016; Liu et al., 2017; Shao et al., 2019), so our knowledge and model treatment of secondary aerosol formation mechanism is still of large uncertainty. To be clearer, the sentences are changed to "Majority of previous model studies underpredict PM concentrations in the north China Plain, especially for aerosol components, such as sulfate, nitrate and SOA concentrations, mainly due to incomplete understanding and treatment of secondary aerosol formation mechanism through multi-phase chemical processes."

*References*

Chen, L., Gao, Y., Zhang, M. G., Fu, J. S., Zhu, J., Liao, H., Li, J. L., Huang, K., Ge, B. Z., Wang, X. M., Lam, Y. F., Lin, C.-Y., Itahashi, S., Nagashima, T., Kajino, M., Yamaji, K., Wang, Z. F., and Kurokawa, J.-I.: MICS-Asia III: multi-model comparison and evaluation of aerosol over East Asia, Atmos. Chem. Phys., 19, 11911–11937, 2019.

Cheng, Y., Zheng, G., Wei, C., Mu, Q., Zheng, B., Wang, Z., Gao, M., Zhang, Q., He, K., Carmichael, G., Poschl, U., and Su, H.: Reactive nitrogen chemistry in aerosol water as a source of sulfate during haze events in China, Sci. Adv., 2, e1601530, https://doi.org/10.1126/sciadv.1601530, 2016.

Gao, M., Han, Z., Liu, Z., Li, M., Xin, J., Tao, Z., Li, J., Kang, J., Huang, K., Dong, X., Zhuang, B., Li, S., Ge, B., Wu, Q., Cheng, Y., Wang, Y., Lee, H., Kim, C., Fu, J. S., Wang, T., Chin, M., Woo, J., Zhang, Q., Wang, Z., and Carmichael G. R.: Air Quality and Climate Change, Topic 3 of the Model Inter-Comparison Study for Asia Phase III (MICS-Asia III), Part I: overview and model evaluation, Atmos. Chem. Phys., 18, 4859–4884, 2018.

Liu, M., Song, Y., Zhou, T., Xu, Z., Yan, C., Zheng, M., Wu, Z., Hu, M., Wu, Y., and Zhu, T.: Fine particle pH during severe haze episodes in northern China, Geophys. Res. Lett., 44, 5213–5221, https://doi.org/10.1002/2017gl073210, 2017.

Shao, J. Y., Chen, Q. J., Wang, Y. X., Lu, X., He, P. Z., Sun, Y. L., Shah, V., Martin, R. V.,

Philip, S., Song, S. J., Zhao, Y., Xie, Z. Q., Zhang, L., and Alexander, B.: Heterogeneous sulfate aerosol formation mechanisms during wintertime Chinese haze events: air quality model assessment using observations of sulfate oxygen isotopes in Beijing, Atmos. Chem. Phys., 19, 6107–6123, 2019.

Wang, G., Zhang, R., Gomez, M. E., Yang, L., Levy Zamora, M., Hu, M., Lin, Y., Peng, J., Guo, S., Meng, J., Li, J., Cheng, C., Hu, T., Ren, Y., Wang, Y., Gao, J., Cao, J., An, Z., Zhou, W., Li, G., Wang, J., Tian, P., Marrero-Ortiz, W., Secrest, J., Du, Z., Zheng, J., Shang, D., Zeng, L., Shao, M., Wang, W., Huang, Y., Wang, Y., Zhu, Y., Li, Y., Hu, J., Pan, B., Cai, L., Cheng, Y., Ji, Y., Zhang, F., Rosenfeld, D., Liss, P. S., Duce, R. A., Kolb, C. E., and Molina, M. J.: Persistent sulfate formation from London Fog to Chinese haze, P. Natl. Acad. Sci. USA, 113, 13630–13635, https://doi.org/10.1073/pnas.1616540113, 2016.

Lines 167-169, please provide references.

**Reply:** The sentences are changed to "An online-coupled regional atmospheric chemistry/aerosol-climate model RIEMS-Chem was used in this study, which was developed based on the Regional Integrated Environmental Model System (RIEMS) (Fu et al., 2005; Wang et al., 2015)." (the two papers have already been included in the manuscript)

Lines 167-180, which microphysical scheme is used in RIEMS-Chem? Does it consider ACI?

**Reply:** We introduce the microphysical scheme in detail above. RIEMS-Chem has considered ACI in this study.

Lines 181-196, Does RIEMS-Chem include aerosol effects on photolysis?

**Reply:** Yes, RIEMS-Chem considers aerosol effects on photolysis by using the TUV model (Lee-Taylor and Madronich, 2007), we add relevant description and reference in the revised version.

*Reference*
Lee-Taylor, J., Madronich, S.: Climatology of UV-A, UV-B, and Erythemal Radiation at the Earth's Surface, 1979-2000, NCAR Technical Note, NCAR/TN-474+STR, pp 1-52. 2007.

Lines 197-208, how does RIEMS-Chem simulate the nitrate formation? Is the organic coating considered in calculation of N2O5 hydrolysis?

**Reply:** $NO_2$, HONO and $HNO_3$ can be produced through gas chemistry and heterogeneous chemistry reactions on aerosol surfaces. $HNO_3$ and $NH_3$ participate in thermodynamic equilibrium processes (represented by ISORROPIA) to form ammonium nitrate. Nitrate can also be formed through heterogeneous reactions on dust and sea salt surfaces. $N_2O_5$ hydrolysis on aqueous aerosols at nighttime is taken into account, but organic coating is not considered yet in calculation of $N_2O_5$ hydrolysis.

Lines 209-226, the authors clarified that they used different size distribution for inorganic, black and organic carbon aerosols. However, they also assumed completely internal mixing

aerosols, which should be represented by the same size distribution. In addition, the aerosol size distribution and mixing state change considerably with development of PM pollution. The authors need at least to include discussions on uncertainties in ARE caused by variations of aerosol size distribution and mixing state.

**Reply:** Thanks for raising this question and we are sorry for the unclear description of aerosol size distribution. In line 216-218 of the previous version, "the geometric mean radius of inorganic, black carbon and organic carbon aerosols were estimated to be 0.1 μm, 0.05 μm and 0.1 μm, with standard deviations of 1.65, 1.6, 1.65, respectively", these are size distributions of each aerosol component measured in clean days, when individual aerosols can be clearly distinguished. In haze days, the time is further classified into light-medium, heavy and severe periods, when about 80% aerosols are measured (with SP2 and SMPS in Beijing) to be internally mixed, with the geometric mean radius of internal mixture being 0.097 μm, 0.11μm and 0.12μm for the three periods, respectively. It is noticed that the mean radius of internal mixture increases slightly with the severity of haze, but in general, the size change is small, so we use a geometric mean radius of 0.11 μm and a standard deviation of 1.65 to represent size distribution of the internal mixing aerosols in this study, which focuses on haze episodes. We add these information in the revised version.

As we introduced above, the measured mixing state of aerosols generally change from external mixing in clean days to internal mixing in haze days. However, the ability of current CTMs in representing evolution of aerosol mixing state is very poor and relevant observations are very limited in China. We assume an internal mixing of anthropogenic aerosols because this study focuses on aerosol feedback effect in haze days with high PM levels. Previous studies indicated that internal mixing of aerosols exerted a stronger positive radiative forcing in the atmosphere and a stronger negative forcing at the surface than those from external mixing (Lesins et al., 2002; Conant et al., 2003). So while the use of internal mixing assumption is reasonable for haze days, it may overestimate the feedback effect for clean days. Sensitivity simulation with respect to size change of internal mixture shows an increase of extinction and absorption coefficients with size within 0.2 μm in the visible light band, and vice versa. So, the use of a constant size distribution for internal mixture could somewhat underestimate and overestimate the feedback effect in severe haze days and light haze days, respectively, but given the small change in the size of internal mixture during haze evolution mentioned above, the effect of size change on aerosol radiative feedback during haze events could be small. We add the above discussions on the uncertainties in ARE and feedback effect caused by variations of aerosol size distribution and mixing state in the conclusions of the revised version.

*Reference*

Conant, W. C., Seinfeld, J. H., Wang, J., Carmichael, G. R., Tang, Y., Uno, I., Flatau, P. J., Markowicz, K. M., and Quinn, P. K.: A model for the radiative forcing during ACE–Asia derived from CIRPAS Twin Otter and R/V Ronald H. Brown data and comparison with observations, J. Geophys. Res. Atmos., 108, 8661, https://doi.org/10.1029/2002JD003260, 2003.

Lesins G., Chylek, P., and Lohmann, U.: A study of internal and external mixing scenarios and its effect on aerosol optical properties and direct radiative forcing, J. Geophys. Res.

Atmos., 107, D10, 10.1029/2001JD000973, 2002.

Lines 227-231, the authors have used ISORROPIA to calculate aerosol water, so it might not be appropriate to consider the hygroscopic growth of inorganic aerosols, BC, POA and SOA.

**Reply:** Sorry for the confusion. Aerosol water from ISORROPIA is just used to determine the value of uptake coefficient of SO2 ($\gamma_{so2}$) on aqueous aerosols. The hygroscopic growth of internal mixture of inorganic aerosols, BC and OA is considered by using the κ-Köhler parameterization. The wet size of aerosol mixture is calculated by Köhler theory with volume weighted hygroscopicity and relative humidity, and the refractive index of wetted aerosols is calculated by Maxwell-Garnett mixing rule (please see the response to the reviewer #1), in which the volume of wetted aerosol is calculated based on the wet aerosol size.

Lines 293-298, the 60km horizontal resolution is too coarse to focus on the Beijing metropolitan.

**Reply:** Thanks for the question. This study investigates the physical and chemical processes of aerosols on a regional scale (the BTH region, most of which are plain). The size of Beijing metropolitan (major urban areas) is about 50km*50km, this study does not intend to look at detailed districts of the Beijing metropolitan, so the 60km horizontal resolution is acceptable for a regional model. And, recently, Tao et al. (2020) examined the impact of model grid resolution on meteorology and air pollutant prediction over the North China Plain (NCP) (most of which is the BTH region) during 2010 using the NASA Unified Weather Research and Forecasting (NU-WRF) model, with horizontal resolutions at 45, 15 and 5 km, respectively. They found that the improvement of air quality modeling was not linear with the resolution increase, the fine resolution did not necessarily predict better results than coarse resolution. e. g. air temperature simulation was not sensitive to the grid resolution, NU-WRF with the 5 km grid simulated the wind speed best, while the 45 km grid yielded the most realistic precipitation as compared to the site observations. Interestingly, for PM$_{2.5}$, the NU-WRF simulation with the 45 km grid generally correlated better with observations than the other two resolutions, and no single resolution gave superior results of MB and RMSE across all sites, e. g. over eight urban and suburban sites in the BTH region, three sites (Baoding, Shuangqing Road and Tanggu) experienced the smallest MB when employing the 5 km resolution grid, two sites (Beijing tower and Longtan Lake) showed the smallest MB using the 15 km grid, while three sites (Xianghe, Tianjin and Tangshan) had the least bias at the 45 km resolution. They explained the important reason weakening the model ability at finer resolutions was the limitation in the resolution (0.25° by 0.25°) of anthropogenic emission inventory (MEIC from Tsinghua University). While projecting the emission inventory at coarse resolution to finer resolution, the representation of emission gradient will be weakened, whereas it could be less affected when projecting emission at finer resolution to coarse resolution by merging grids, so the selection of model grid resolution should be consistent with the available resolution of emission inventory.

Our selection of 60 km resolution is also a compromise of simulation accuracy and computational cost because the online coupled model accounting for interactions between physics/dynamic and chemistry and the process analysis tracing numerous species and

processes at each time step are very computationally expensive, so please understand our choice for this model grid resolution for this study.

Given the generally good ability of the model in reproducing spatial distribution and temporal variation of meteorological variables, boundary layer height, aerosol compositions, as well as aerosol optical properties, the model results and main conclusions from this study would be reasonable and reliable. We are aware that finer grid resolution may better represent characteristics at urban scale, we would like to use finer model grid resolution along with emission inventory with finer resolution, availability of vertical observation and higher computational efficiency in the future.

We add discussions on potential uncertainties in model grid resolution, assumptions for mixing state and size distribution in the conclusions of the revised version as:

"This study is still subject to some uncertainties: 1.) An internal mixing was assumed for aerosol mixing in this study, but the mixing state of aerosols is always changing, while this assumption is generally realistic for haze days, it may overestimate the feedback effect for clean days. 2.) A typical size distribution measured during haze days was used, whereas the size of aerosol internal mixture could change to some extent with aging processes. These uncertainties require further development of model treatment for evolution of aerosol mixing state and size distribution, which is poorly represented in current online coupled models. 3.) Direct aerosol radiative effect dominated the feedback effect in this study, so more cases in different regions and seasons, when indirect effect could be more important are needed to elucidate the complete feedback mechanism at different spatial and temporal scales. 4.) Finer model grid resolution is expected to be applied to look into details of the feedback effect at urban scale along with emission inventory at finer resolution, vertical observations and higher computational efficiency when available in the future."

***Thanks again for the comments and suggestions***

Next is the supplement for additional figures and tables

**Supplement to "Aerosol radiative effects and feedbacks on boundary layer meteorology and PM2.5 chemical components during winter haze events over the Beijing-Tianjin-Hebei region"**

Jiawei Li[1], Zhiwei Han[*1,2], Yunfei Wu[1], Zhe Xiong[1], Xiangao Xia[3], Jie Li[1,2], Lin Liang[1,2], Renjian Zhang[1]

[1] Key Laboratory of Regional Climate-Environment for Temperate East Asia, Institute of Atmospheric Physics, Chinese Academy of Sciences, Beijing 100029, China
[2] University of Chinese Academy of Sciences, Beijing 100049, China
[3] Key Laboratory of Middle Atmosphere and Global Environment Observation, Institute of Atmospheric Physics, Chinese Academy of Sciences,
Beijing100029, China
Correspondence to: Zhiwei Han (hzw@mail.iap.ac.cn)

[Figure]

Figure S1. Period mean vertical profiles of temperature, RH, and wind speed from sounding data at the Beijing Observatory and corresponding model results for the haze episode 1 (20-26 February). The soundings were conducted at 8:00 and 20:00 of local standard time (LST=UTC+8).

[Figure]

Figure S2. Same as Figure S1 but for the haze episode 2 (1-4 March).

[Figure]

Figure S3. Observed and simulated hourly $O_3$, $SO_2$, $NO_2$, and $PM_{2.5}$ concentrations in Beijing. Observations were obtained from the CNEMC (China National Environmental Monitoring Center, http://www.cnemc.cn/). The observations are averages of the 12 measurement sites in Beijing (n=12).

[Figure]

Figure S4. Same as Figure S3 but for the Tianjin city (n=15).

[Figure]

Figure S5. Same as Figure S3 but for the Baoding city (n=6).

[Figure]

Figure S6. Same as Figure S3 but for the Shijiazhuang city (n=8).

[Figure]

Figure S7. Same as Figure S3 but for Handan (n=4).

[Figure]

Figure S8. Same as Figure S3 but for the Chengde city (n=5).

Table S1. Performance statistics for meteorology soundings at the Beijing Observatory for the haze episode 1 (20-26 February). Averages of wind speed (WS), temperature (T), relative humidity (RH), and mixing ratio (Q) from observation (Obs) and model simulation (Sim) as well as correlation coefficients (R) are given.

| | WS (m s$^{-1}$) | T (℃) | RH (%) | Q (g kg$^{-1}$) |
|---|---|---|---|---|
| **Troposphere** (sample number = 469) | | | | |
| Obs | 11.3 | -20.9 | 53.2 | 1.6 |
| Sim | 11.0 | -19.5 | 59.2 | 1.6 |
| R | 0.97 | 0.997 | 0.74 | 0.95 |
| **Below 3km** (sample number = 204) | | | | |
| Obs | 5.4 | -2.1 | 73.1 | 2.8 |
| Sim | 5.7 | -1.4 | 75.3 | 3.0 |
| R | 0.68 | 0.937 | 0.64 | 0.78 |

Table S2. Same as Table S1 but for the haze episode 2 (1-4 March).

| | WS (m s$^{-1}$) | T (℃) | RH (%) | Q (g kg$^{-1}$) |
|---|---|---|---|---|
| **Troposphere** (sample number = 228) | | | | |
| Obs | 24.7 | -23.2 | 36.9 | 0.8 |
| Sim | 23.5 | -21.9 | 42.3 | 0.9 |
| R | 0.99 | 0.996 | 0.72 | 0.93 |
| **Below 3km** (sample number = 98) | | | | |
| Obs | 7.6 | -2.5 | 42.0 | 1.6 |
| Sim | 7.1 | -1.9 | 43.1 | 1.7 |
| R | 0.72 | 0.940 | 0.73 | 0.87 |

Table S3. Performance statistics for hourly $O_3$, $SO_2$, $NO_2$, and $PM_{2.5}$ concentrations in 13 cities of the BTH region for the study period (17 February - 12 March). Mean observation (Obs, unit=µg $m^{-3}$), mean simulation (Sim, unit=µg $m^{-3}$), correlation coefficient (R), and normalized mean bias (NMB, unit=%) are presented.

| City[a] | $PM_{2.5}$ | | | | $O_3$ | | | | $SO_2$ | | | | $NO_2$ | | | |
|---|---|---|---|---|---|---|---|---|---|---|---|---|---|---|---|---|
| | Obs | Sim | R | NMB | Obs | Sim | R | NMB | Obs | Sim | R | NMB | Obs | Sim | R | NMB |
| BJ | 125.4 | 128.2 | 0.8 | 2 | 29.8 | 29.4 | 0.74 | -1 | 57.7 | 64.1 | 0.51 | 11 | 73.2 | 66.7 | 0.66 | -9 |
| TJ | 113.7 | 107.8 | 0.75 | -5 | 25.8 | 22.1 | 0.72 | -14 | 100.6 | 133 | 0.52 | 32 | 74.3 | 67.3 | 0.62 | -9 |
| SJZ | 211.7 | 219.3 | 0.65 | 4 | 26.7 | 26.6 | 0.63 | 0 | 103.8 | 93.9 | 0.25 | -10 | 77.4 | 73.7 | 0.42 | -5 |
| TS | 150.9 | 234.8 | 0.78 | 56 | 29.8 | 27.7 | 0.7 | -7 | 148.1 | 131.2 | 0.62 | -11 | 71.5 | 120.4 | 0.6 | 68 |
| QHD | 93.3 | 100.8 | 0.81 | 8 | 41.6 | 27.8 | 0.7 | -33 | 71.8 | 48.8 | 0.62 | -32 | 46.6 | 43 | 0.76 | -8 |
| HD | 184.7 | 130.1 | 0.76 | -30 | 28.6 | 23.6 | 0.61 | -17 | 80.8 | 59.1 | 0.55 | -27 | 78.1 | 68.6 | 0.62 | -12 |
| BD | 172.7 | 140.4 | 0.74 | -19 | 27.4 | 21.4 | 0.7 | -22 | 117.3 | 98.9 | 0.47 | -16 | 86.3 | 85.7 | 0.42 | -1 |
| ZJK | 109.6 | 99.6 | 0.81 | -9 | 41.5 | 50.9 | 0.65 | 23 | 107.5 | 26.2 | 0.41 | -76 | 51.2 | 29.5 | 0.74 | -42 |
| CD | 98.3 | 96.8 | 0.84 | -2 | 34.9 | 32.9 | 0.77 | -6 | 56.7 | 55.1 | 0.18 | -3 | 47.6 | 43.1 | 0.73 | -9 |
| LF | 152.1 | 140.4 | 0.78 | -8 | 27.8 | 28.6 | 0.71 | 3 | 73.6 | 60.1 | 0.52 | -18 | 76.3 | 68.7 | 0.57 | -10 |
| CZ | 125.9 | 119.2 | 0.72 | -5 | 44.8 | 38.6 | 0.4 | -14 | 66.5 | 74 | 0.17 | 11 | 35.7 | 75.5 | 0.51 | 111 |
| HS | 149.7 | 136.6 | 0.71 | -9 | 41.6 | 20.6 | 0.63 | -50 | 72.2 | 54.7 | 0.23 | -24 | 54.6 | 34.1 | 0.53 | -38 |
| XT | 230.9 | 258.4 | 0.71 | 12 | 32.7 | 35.7 | 0.51 | 9 | 131.5 | 91.8 | 0.37 | -30 | 82.3 | 82.8 | 0.4 | 1 |
| Total | 147.9 | 147.3 | 0.87 | -0.4 | 33.4 | 29.7 | 0.81 | -11 | 91.5 | 76.3 | 0.60 | -17 | 65.8 | 66.1 | 0.74 | 0.5 |

a: Names of the cities, the numbers of the monitoring sites in each city (n), and the latitudes and longitudes of the cities are:

  BJ = Beijing (n=12), 40.0°N and 116.4°E;

  TJ = Tianjin (n=15), 39.1°N and 117.2°E;

  SJZ = Shijiazhuang (n=8), 38.0°N and 114.5°E;

  TS = Tangshan (n=6), 39.6°N and 118.2°E;

  QHD = Qinhuangdao (n=5), 39.9°N and 119.6°E;

  HD = Handan (n=4), 36.6°N and 114.5°E;

  BD = Baoding (n=6), 38.9°N and 115.5°E;

  ZJK = Zhangjiakou (n=5), 40.8°N and 114.9°E;

  CD = Chengde (n=5), 41.0°N and 117.9°E;

  LF = Langfang (n=4), 39.5°N and 116.7°E;

  CZ = Cangzhou (n=3), 38.3°N and 116.8°E;

  HS = Hengshui (n=3), 37.7°N and 115.7°E;

  XT = Xingtai (n=4), 37.0°N and 114.5°E.

---

## Editor Decision (ED1)

The line numbers below are based on the change-marked version of the revised manuscript.

L97: Please give the full name of PM.

L180-181: Please rewrite this sentence. It should be "transport of pollutants are driven....., and changes of .... provide feedback to ....".

L183: "chemistry".

L204-206: Is it aerosol liquid water content (ALWC)?

L225-228: Are these radius values for dry or wet aerosols?

L279: Remove the word "following" since land surface module and boundary layer module are not described in the following paragraph.

L280: change "account for" to "calculate"

L286-287: It is difficult to understand how the coupling works in the model from this sentence. What was the integration time step of the base model (RIEMS)?At every 2.5 minutes or 30 minutes were the physical variables, e.g., T and J-values, updated for chemical calculations? Was the feedback allowed to occur at every 30 minutes?

L292: "has" ?

L299: The species concentrations are affected by emissions, dynamic transport, and physical and chemical process.

L310: Cloud chemistry should be included in the chemical processes although it is generally taken into account in the cloud module.
What the physical and chemical processes refer to should be clearly clarified from the beginning and used correctly throughout the entire manuscript. It may not be appropriate to consider advection and diffusion as physical processes, which are different from real physical processes affecting aerosols such as condensation and coagulation.

L357-359: The definition of the NoAer simulation is not well described, which was also mentioned by Referee#1. It is stated in Line 208-209 that RIEMS-Chem treats 9 aerosol types including sulfate, nitrate, ammonium, black carbon (BC), primary organic aerosol (POA), secondary organic aerosol (SOA), anthropogenic primary PMs

($PM_{2.5}$ and $PM_{10}$), dust and sea salt. What did you do with these types of aerosols to remove anthropogenic ones considering that some type of aerosols can originate from both natural and anthropogenic sources? It might be more appropriate to use NoAerfeedback or NoFB for this simulation.

L596: Is this section title appropriate considering that Sect. 5 also presents model results?

L805: Again, It might not be suitable to consider emissions as physical process.

L855, L903 and L908: How or what can gas-phase chemistry lead to formation of $PM_{2.5}$? The reaction of $SO_2$ with OH in the gas phase produces $H_2SO_4$, which is acid gas not sulfate, and need to undergo nucleation or condensation to form sulfate aerosols. Are the formation rates of aerosols (including those in the clean stage stated in Line 843) comparable to the results from previous studies?

L1224-1225: Please rewrite or delete this sentence.
L1226: Specify what "aerosol effects" refer to or remove "with and without aerosol effects"

L1325-1327:It is the size distribution pattern, more specifically the distribution shape parameters, was used. See my comments on L225-228 and L1224-1225. Some descriptions in the first paragraph of the conclusions are repeated here. The conclusions can be more concise.

---

## Author Response (AR2)

Dear Editor,

Thank you very much for the valuable comments and careful reading which improve the manuscript. The responses (blue font) are as follows:

The line numbers below are based on the change-marked version of the revised manuscript.

L97: Please give the full name of PM.
**Reply:** Revised.

L180-181: Please rewrite this sentence. It should be "transport of pollutants are driven….., and changes of …. provide feedback to ….".
**Reply:** Revised to "Transport of pollutants are driven by meteorological fields from RIEMS and changes of pollutants exert feedbacks on the dynamic and physical modules".

L183: "chemistry".
**Reply:** Revised.

L204-206: Is it aerosol liquid water content (ALWC)?
**Reply:** Yes, revised to "aerosol liquid water content (ALWC)".

L225-228: Are these radius values for dry or wet aerosols?
**Reply:** These radius values are for dry aerosols.

L279: Remove the word "following" since land surface module and boundary layer module are not described in the following paragraph.
**Reply:** "following" changed to "subsequently called".

L280: change "account for" to "calculate"
**Reply:** Revised.

L286-287: It is difficult to understand how the coupling works in the model from this sentence. What was the integration time step of the base model (RIEMS)?At every 2.5 minutes or 30 minutes were the physical variables, e.g., T and J-values, updated for chemical calculations? Was the feedback allowed to occur at every 30 minutes?
**Reply:** Sorry for the confusion. It is revised to "All the physical modules are called every 2.5 minutes and the transfer of variables between chemical module and radiation/meteorological modules is made every 30 minutes". So, the feedback was allowed to occur at every 30 minutes

L292: "has" ?
**Reply:** Revised.

L299: The species concentrations are affected by emissions, dynamic transport, and physical and chemical process.
**Reply:** Revised.

L310: Cloud chemistry should be included in the chemical processes although it is generally taken into account in the cloud module.
What the physical and chemical processes refer to should be clearly clarified from the beginning and used correctly throughout the entire manuscript. It may not be appropriate to consider advection and diffusion as physical processes, which are different from real physical processes affecting aerosols such as condensation and coagulation.
**Reply:** The description is revised to "chemical processes (gas-phase chemistry, aqueous chemistry, thermodynamic equilibrium and heterogeneous reactions)", but the contribution of aqueous chemistry is very small during the study period. It's difficult to clearly distinguish between physical process and dynamic process, but compared with chemical process, it is acceptable to consider "advection and diffusion" as physical processes, and in the process analysis, for clarity and easy understanding, we classify all the processes into two categories: physical and chemical processes, physical processes refer to all the processes (including emission) except chemistry. It might be odd and confused to use three categories as "dynamical, physical and chemical processes", so we would like to keep the current classification.

L357-359: The definition of the NoAer simulation is not well described, which was also mentioned by Referee#1. It is stated in Line 208-209 that RIEMS-Chem treats 9 aerosol types including sulfate, nitrate, ammonium, black carbon (BC), primary organic aerosol (POA), secondary organic aerosol (SOA), anthropogenic primary PMs ($PM_{2.5}$ and $PM_{10}$), dust and sea salt. What did you do with these types of aerosols to remove anthropogenic ones considering that some type of aerosols can originate from both natural and anthropogenic sources? It might be more appropriate to use NoAerfeedback or NoFB for this simulation.
**Reply:** Yes, thanks for the good suggestion, actually, in NoAer, we inactive the feedback of aerosols, NoAer could be misleading, so we change "NoAer" to "NoAFB" throughout the revised version.

L596: Is this section title appropriate considering that Sect. 5 also presents model results?
**Reply:** Yes, the title is not appropriate, so it is revised to "Aerosol radiative effects and feedbacks"

L805: Again, It might not be suitable to consider emissions as physical process.

**Reply:** As we explained above, it is acceptable to consider emission release as physical process for brevity.

L855, L903 and L908: How or what can gas-phase chemistry lead to formation of $PM_{2.5}$? The reaction of $SO_2$ with OH in the gas phase produces $H_2SO_4$, which is acid gas not sulfate, and need to undergo nucleation or condensation to form sulfate aerosols. Are the formation rates of aerosols (including those in the clean stage stated in Line 843) comparable to the results from previous studies?

**Reply:** Yes, sulfate is formed through several pathways: one of them is the gas-phase oxidation of $SO_2$ by OH to form sulfuric acid ($H_2SO_4$), followed by nucleation or condensation into particulate phase, which is normally regarded as gas chemistry process (Seinfeld and Pandis, 2006; Cheng et al., 2016), so we assign this process to gas chemistry (GAS). So far, the direct field measurements of formation rates of secondary aerosols in this region is not available for this study, but the model simulations for secondary aerosol concentrations have been compared with daily mean observations in Figure 3, which shows a generally good agreement. It needs to be explained here that about 3% of $SO_2$ emission is assumed to be sulfuric acid, which transform entirely and rapidly to sulfate, and this process is assigned to gas chemistry as well. We clarify the definition of GAS as "the gas-phase oxidation of $SO_2$ by OH to form sulfuric acid ($H_2SO_4$), followed by nucleation or condensation into particulate phase" in the revised version.

Cheng, Y., Zheng, G., Wei, C., Mu, Q., Zheng, B., Wang, Z., Gao, M., Zhang, Q., He, K., Carmichael, G., Poschl, U., and Su, H.: Reactive nitrogen chemistry in aerosol water as a source of sulfate during haze events in China, Sci. Adv., 2, e1601530, https://doi.org/10.1126/sciadv.1601530, 2016.

Seinfeld J. H., Pandis S. N.: Atmospheric Chemistry and Physics: From Air Pollution to Climate Change, John Wiley & Sons, Hoboken, NJ, 2006.

L1224-1225: Please rewrite or delete this sentence.
**Reply:** Deleted.

L1226: Specify what "aerosol effects" refer to or remove "with and without aerosol effects"
**Reply:** We remove "with and without aerosol effects".

L1325-1327:It is the size distribution pattern, more specifically the distribution shape parameters, was used. See my comments on L225-228 and L1224-1225. Some descriptions in the first paragraph of the conclusions are repeated here. The conclusions can be more concise.
**Reply:** We delete the description in L1224-1225 to be more concise.